# Measured greenhouse gas budgets challenge emission savings from palm-oil biodiesel

Ana Meijide [1,2,3 ✉], Cristina de la Rua [4], Thomas Guillaume [5,6,7], Alexander Röll [8], Evelyn Hassler[9], Christian Stiegler [1], Aiyen Tjoa[10], Tania June[11], Marife D. Corre[9], Edzo Veldkamp [9,12] & Alexander Knohl [1,12]

The potential of palm-oil biofuels to reduce greenhouse gas (GHG) emissions compared with fossil fuels is increasingly questioned. So far, no measurement-based GHG budgets were available, and plantation age was ignored in Life Cycle Analyses (LCA). Here, we conduct LCA based on measured $CO_2$, $CH_4$ and $N_2O$ fluxes in young and mature Indonesian oil palm plantations. $CO_2$ dominates the on-site GHG budgets. The young plantation is a carbon source ($1012 \pm 51$ gC m$^{-2}$ yr$^{-1}$), the mature plantation a sink ($-754 \pm 38$ gC m$^{-2}$ yr$^{-1}$). LCA considering the measured fluxes shows higher GHG emissions for palm-oil biodiesel than traditional LCA assuming carbon neutrality. Plantation rotation-cycle extension and earlier-yielding varieties potentially decrease GHG emissions. Due to the high emissions associated with forest conversion to oil palm, our results indicate that only biodiesel from second rotation-cycle plantations or plantations established on degraded land has the potential for pronounced GHG emission savings.

[1] Bioclimatology, University of Göttingen, Büsgenweg 2, 37077 Göttingen, Germany. [2] Department of Crop Sciences, Division Agronomy, University of Göttingen, Von Siebold Str. 8, 37075 Göttingen, Germany. [3] Ecology, University of Granada, Avenida Fuente Nueva s/n, 18071 Granada, Spain. [4] Department of Electrical and Computer Engineering, Renewable and Sustainable Energy Systems, Technical University of Munich, Lichtenbergstraße 4a, 85748 Garching, München, Germany. [5] Soil Science of Temperate Ecosystems, University of Göttingen, Büsgenweg 2, 37077 Göttingen, Germany. [6] School of Architecture, Civil and Environmental Engineering (ENAC), Ecole Polytechnique Fédérale de Lausanne (EPFL), Ecological Systems Laboratory (ECOS), Station 2, Lausanne 1015, Switzerland. [7] Swiss Federal Institute for Forest, Snow and Landscape Research (WSL), Site Lausanne, Station 2, Lausanne 1015, Switzerland. [8] Tropical Silviculture and Forest Ecology, University of Göttingen, Büsgenweg 1, 37077 Göttingen, Germany. [9] Soil Science of Tropical and Subtropical Ecosystems, University of Göttingen, Büsgenweg 2, 37077 Göttingen, Germany. [10] Fakultas Pertanian, Universitas Tadulako, Palu, Sulawesi, Indonesia. [11] IPB University, Department of Geophysics and Meteorology, Bogor, Indonesia. [12] Center of Biodiversity and Sustainable Land Use, University of Göttingen, Büsgenweg 1, Göttingen 37077, Germany. ✉email: ana.meijideorive@uni-goettingen.de

Vegetable oil-based biofuels are increasingly used as an alternative to fossil fuels. Initially considered climate-friendly substitutes that improve the greenhouse gas (GHG) budget of the transport sector[1,2], they were politically endorsed. Producing countries, such as Indonesia, continue to promote their use in order to decrease their dependence on external fossil fuels. The European Union (EU) recently weakened the endorsement of palm-oil biofuels due to, among other reasons, controversies regarding their climate benefits. However, the EU consumed more that 7 million tons of palm oil in 2018, of which more than half was used for biodiesel[3]. Overall, this has stimulated the global palm-oil demand and is driving the ongoing expansion of oil palm (*Elaeis guineensis Jacq.*) across the tropics[4]. Indonesia currently accounts for about half of the world's palm-oil production[5], with 10.27 million ha covered by oil palms in 2017[5]. The palm-oil production is centred in Sumatra and Kalimantan[6], where substantial forest losses and land-use changes have occurred over the last decades and are still ongoing[7]. Previous studies have shown that conversion of rainforest to produce food crop-based biofuels and particularly palm-oil biodiesel releases substantially more GHG than the biofuels save by displacing fossil fuels[8–12]. This is mainly caused by the large carbon (C) emissions after forest conversion[8,13]. While first ecosystem-scale carbon dioxide ($CO_2$) measurements point to substantial C uptake in oil palm plantations[14,15], there so far is no information on annual $CO_2$ budgets at the ecosystem level, and previous soil-based $CO_2$ flux measurements are limited[16–18]. There thus is an urgent need for field-based, ecosystem-scale measurements of C fluxes in oil palm plantations to comprehensively assess the GHG balance of palm-oil biodiesel.

Conversion of forests to agricultural plantations may further contribute substantially to non-$CO_2$ GHG emissions[19], but field quantifications of methane ($CH_4$, global warming potential, GWP of 25[2,20]) and nitrous oxide ($N_2O$, GWP of 298[2,20]) emissions in oil palm plantations are so far limited[16–18,21,22]. The net global warming potential ($GWP_{net}$)[23], which includes the net ecosystem exchange (NEE) of $CO_2$ fluxes as well as soil $CH_4$ and $N_2O$ fluxes, has previously not been assessed for oil palm plantations, despite their rapid expansion across the tropics. Available field-based studies on oil palms revealed large increases in leaf area index[24] and transpiration[25] with increasing plantation age and thus palm dimensions. As we expect a corresponding increase in C uptake with oil palm age, comprehensive GHG budgets should consider different stages of oil palm development. This need is further emphasized by the considerable variability in management practices (e.g., regarding fertilization, weeding) that are employed at different plantation development stages[15,21].

The sustainability of palm-oil biodiesel remains actively debated[26], not only with regard to GHG emissions but also considering other ecosystem properties and functions such as biodiversity[27], microclimate[28,29] and C pools and fluxes[30,31]. Life cycle analyses (LCA) are a tool to quantify the climatic impacts of palm-oil-derived biofuels[32–34]. Previous LCA studies suggest that replacement of fossil fuel with biofuel can lead to contrasting GHG savings[32–36]. As such, GHG savings can be limited if biofuel production leads to substantial land-use change-related emissions[8,9,37]. This led the European Union (EU) to define minimum GHG savings requirements for biofuels compared with fossil fuel in its renewable energy directive[2]: LCA has to show at least 60% GHG-emission savings for biofuels that start production operation before 2021, at least 65% for operations starting between 2021 and 2025 and at least 80% for operations after 2026. Therein, the directive and most previous studies consider biofuels to be $CO_2$ neutral, i.e., the $CO_2$ absorbed during cultivation is released during combustion (hereafter referred to as 'C neutrality assumption'[38]. Some previous studies have considered biogenic $CO_2$ emissions during the plantation life cycle by analysing changes in aboveground biomass[13,38,39], but unfortunately these results have not yet been incorporated into LCA for palm-oil biodiesel. In our study, we analyse the entire life cycle of palm-oil biodiesel, which for the first time includes field-based GHG measurements during different stages of oil palm cultivation. We show that oil palm cultivation in the first rotation cycle after forest conversion leads to no GHG savings from palm-oil biofuel compared with fossil fuel. Only biodiesel from second-rotation-cycle plantations or from plantations established on degraded land can lead to the required GHG-emission savings. We propose alternative management scenarios that can potentially increase GHG-emission savings.

## Results

**Larger GHG emissions in young plantations and organic soils.** The NEE of $CO_2$ indicates that the 1-year-old plantation on mineral soil was a large C source ($1012 \pm 51$ gC m$^{-2}$ yr$^{-1}$; Fig. 1; Supplementary Table 1). In contrast, the 12-year-old plantation on mineral soil was a considerable on-site C sink ($-799 \pm 40$ gC m$^{-2}$ yr$^{-1}$ for the first year and $-709 \pm 35$ gC m$^{-2}$ yr$^{-1}$ for the second year; Supplementary Table 1). However, in terms of net ecosystem productivity (NEP = NEE + harvest C export) even

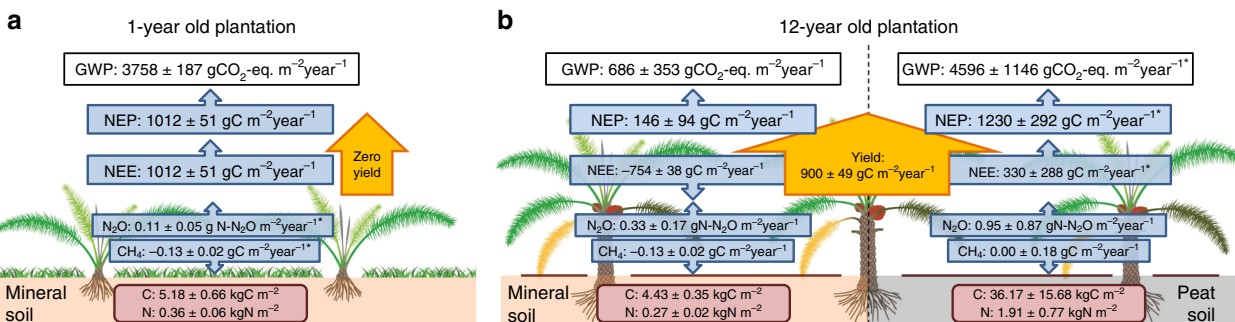

**Fig. 1 Measured greenhouse gases at the oil palm plantations.** Ecosystem greenhouse gases measured at the young (1-year-old; **a**) and mature (12-year-old; **b**) oil palm plantations on mineral soils (-**b**, left) and peat soils (-**b**, right). Greenhouse gas fluxes (blue boxes) of carbon dioxide, i.e., net ecosystem exchange (NEE) measured at the ecosystem level using the eddy covariance technique, and of soil methane ($CH_4$) and nitrous oxide ($N_2O$) fluxes measured using vented static chambers. Net ecosystem productivity (NEP) is estimated as the sum of yield (which is zero in case of the young plantation) and NEE. Further presented are soil organic carbon and nitrogen stocks in the top 30 cm of the soil (red boxes) and the ecosystem net global warming potential (GWP, black boxes). Negative and positive fluxes indicate uptake and emission, respectively. See Supplementary Tables 1 and 2 for additional details on all presented fluxes. * indicates that fluxes were not directly measured in our study sites, but estimated from other measurements.

the 12-year-old plantation was a C source with considerable inter-annual variability (52–240 gC m$^{-2}$ yr$^{-1}$), possibly caused by the strong El Nino year during our measurement period[28,40,41]. The 1-year-old plantation, not having produced fruits yet, had no C exported via harvest, and NEP thus equalled NEE.

Diurnal NEE fluxes (Supplementary Fig. 1) show that during the night, when no photosynthesis occurred, ecosystem respiration was similar in both plantations, despite of the much smaller size of palms in the young plantation. These similar night-time C emissions suggest that the lower autotrophic respiration by smaller oil palms in the 1-year plantation was probably substantially enhanced by heterotrophic soil respiration and by autotrophic respiration of the understory vegetation (annual crops and grasses, only in the young plantation[24,25]; herbicide weed control in the mature plantation)[15]. During the day, the large photosynthesis rates needed to produce fruits resulted in much larger NEE and net $CO_2$ uptake in the 12- than in the 1-year-old plantation (Supplementary Fig. 1).

Measured soil respiration was similar from mineral soil in the 1- and 12-year-old plantations ($P = 0.1838$; linear mixed effect models), with values of $133.9 \pm 56.9$ and $91.7 \pm 39.6$ mgC m$^{-2}$ h$^{-1}$, respectively (mean ± SE, $n = 3$ and 4; Supplementary Fig. 2a). Autotrophic respiration is expected to be lower in the 1-year-old plantation due to the lower size of palms, suggesting that soil organic carbon loss by heterotrophic respiration was higher in the 1-year-old plantation and supporting previous studies on soil C stock decay over the plantation life cycle[31,42] (also see Supplementary Fig. 3).

As previously reported for oil palm plantations in the area[16–18], both plantations were small $CH_4$ sinks (mean fluxes of $-17.0 \pm 2.2$ and $-14.7 \pm 4.6$ mgC m$^{-2}$ h$^{-1}$ for the 1- and 12-year-old plantations, respectively; Supplementary Fig. 2b). As the 1-year-old plantation was not fertilized during our measurement period, we did not detect any pulses of soil $N_2O$ emissions (Supplementary Fig. 2c). In contrast, we observed several peaks of soil $N_2O$ emissions from the 12-year-old plantation (with mean daily $N_2O$ emissions of up to $296 \pm 279$ µgN$_2$O–N m$^{-2}$ h$^{-1}$) associated with fertilizer application (Supplementary Fig. 2c). Annual soil $N_2O$ emissions were $0.3 \pm 0.17$ gN$_2$O–N m$^{-2}$ yr$^{-1}$, which is almost three times higher than from smallholder oil palm plantations in the area (Supplementary Table 2), which have on average two-times lower fertilization rates than our mature plantation site[21]. In our study, we used the annual $N_2O$ emissions measured in the mentioned smallholder plantations ($0.11 \pm 0.05$ µgN$_2$O–N m$^{-2}$ h$^{-1}$; Supplementary Table 2)[21] for the 1-year-old plantation, as our measurements were not enough to report annual estimates. Emissions from oil palm canopy soil, i.e., the soil lodged between the stems and leaf axils of oil palms, were neglected, as they contribute only 1% to the total $N_2O$ fluxes and 0.2% to total $CH_4$ emissions[43].

Our results show that the GWP$_{net}$ of oil palm plantations on mineral soils is dominated by $CO_2$ fluxes, with only minor contributions from soil $CH_4$ and $N_2O$ emissions (Fig. 1). The strongly positive NEE of the young plantation contrasted that of the mature plantation (strongly negative NEE), which highlights the dynamic changes in $CO_2$ fluxes of oil palm plantations from establishment to maturity (Supplementary Table 1). When including the C exports through harvest into the C balance (i.e., using the NEP), even the mature plantation was a net C source (Fig. 1). We deduce that the high GWP$_{net}$ and NEP (Supplementary Tables 1 and 2) from first rotation-cycle oil palm plantations after forest conversion originate from enhanced decomposition of soil organic carbon and reduced C input after conversion to agricultural land[30,31,44,45]. Our study provides a first measurement-based confirmation that first rotation-cycle oil palm plantations, despite their large C uptake rates during mature stages, are net C emitters when all ecosystem-level components of the C budget are included and when different stages of oil palm cultivation are considered (Fig. 1). Oil palm plantations are a net source of $CO_2$, with a greenhouse gas intensity (GHGI[23] = GWP in $CO_2$-equivalents ($CO_2$-eq.) ÷ crop yield) between 0.1 and 0.4 gCO$_2$ g$^{-1}$yield.

Soil GHG emissions from organic soils are larger than from mineral soils[46], and especially nitrogen-rich organic soils in the tropics are global $N_2O$ hotspots[47]. Although Indonesia signed a moratorium on the conversion of forest and peatlands[48] in order to reduce the country's GHG emissions, many existing plantations are located on organic soils or include areas with peat soils. Until now, soil GHG fluxes from oil palm plantations located on mineral vs. organic soils have not been compared. We measured soil respiration, $N_2O$ and $CH_4$ fluxes from mineral and peat soils in the mature plantation and observed 2.6-times larger soil respiration from the peat soils ($P = 0.003$; linear mixed effects model; Supplementary Fig. 2; Supplementary Table 3). $N_2O$ and $CH_4$ fluxes from peat soils were also much larger: $N_2O$ fluxes from peat soils were about three times larger ($0.33 \pm 0.17$ and $0.95 \pm 0.87$ gN$_2$O–N m$^{-2}$ yr$^{-1}$ for mineral and peat soils, respectively), and mineral soils acted as a $CH_4$ sink ($-0.13 \pm 0.02$ gC m$^{-2}$ yr$^{-1}$) while peat soils were sometimes a $CH_4$ source ($0.00 \pm 0.18$ gC m$^{-2}$ yr$^{-1}$; Supplementary Fig. 2; Fig. 1). Due to the large variability of the fluxes, we did not find significant differences with mineral soils in the temporal datasets ($P = 0.098$ and 0.735 for $N_2O$ and $CH_4$, respectively), but the differences in means were significant ($P < 0.001$; Mann–Whitney $U$ test; Supplementary Table 3). We observed generally low GHG emissions during dry seasons (e.g., July to October 2015, when intense drought occurred due to a strong El Nino event[28,40,41]; Supplementary Fig. 2), and pulses of soil $N_2O$ emission following fertilization. Our results are in line with a previous report of very large $N_2O$ emissions from oil palm plantations on peat[49].

We estimated ecosystem-scale NEE for mature oil palm plantations on peat by using our NEE measurements on mineral soils (Supplementary Table 1), subtracting measured soil respiration on mineral soils and adding measured soil respiration on peat (Supplementary Table 2). Our estimations show that as opposed to mature plantations on mineral soils, which at the ecosystem-scale were a large C sink, mature plantations on peat soils were a C source (NEE = $330 \pm 288$ gC m$^{-2}$ yr$^{-1}$; Fig. 1). Ecosystem GWP$_{net}$ was about seven times larger for mature plantations on peat than for mature plantations on mineral soils. In our study, peat soils cover only 5–10% of the total area of the mature plantation; when peat is more dominant or a plantation is entirely located on organic soils, emissions from palm-oil biodiesel are expected to increase substantially, in accordance with the larger GWP$_{net}$ (Fig. 1; Supplementary Table 3). We do not have any measurements in young plantations on peat, where due to the high soil C contents soil respiration is expected to be very large. We therefore refrain from providing a full LCA for palm-oil biodiesel from plantations on peat soils, and solely focus on mineral soils in the subsequent analysis.

**Enhanced palm-oil biodiesel LCA challenges emission savings.** For comparison, we first performed an LCA following the traditional, commonly applied approach, which assumes C neutrality. The $CO_2$ captured by the ecosystem during production is considered to be equal to the emissions during combustion, and both are disregarded in the analysis[38,50] (hereafter referred to as traditional LCA). According to the traditional LCA, 1 MJ of biodiesel from palm oil leads to emissions of 186 gCO$_2$-eq. (173–199 gCO$_2$-eq. MJ$^{-1}$, 25- and 75 percentiles from Monte

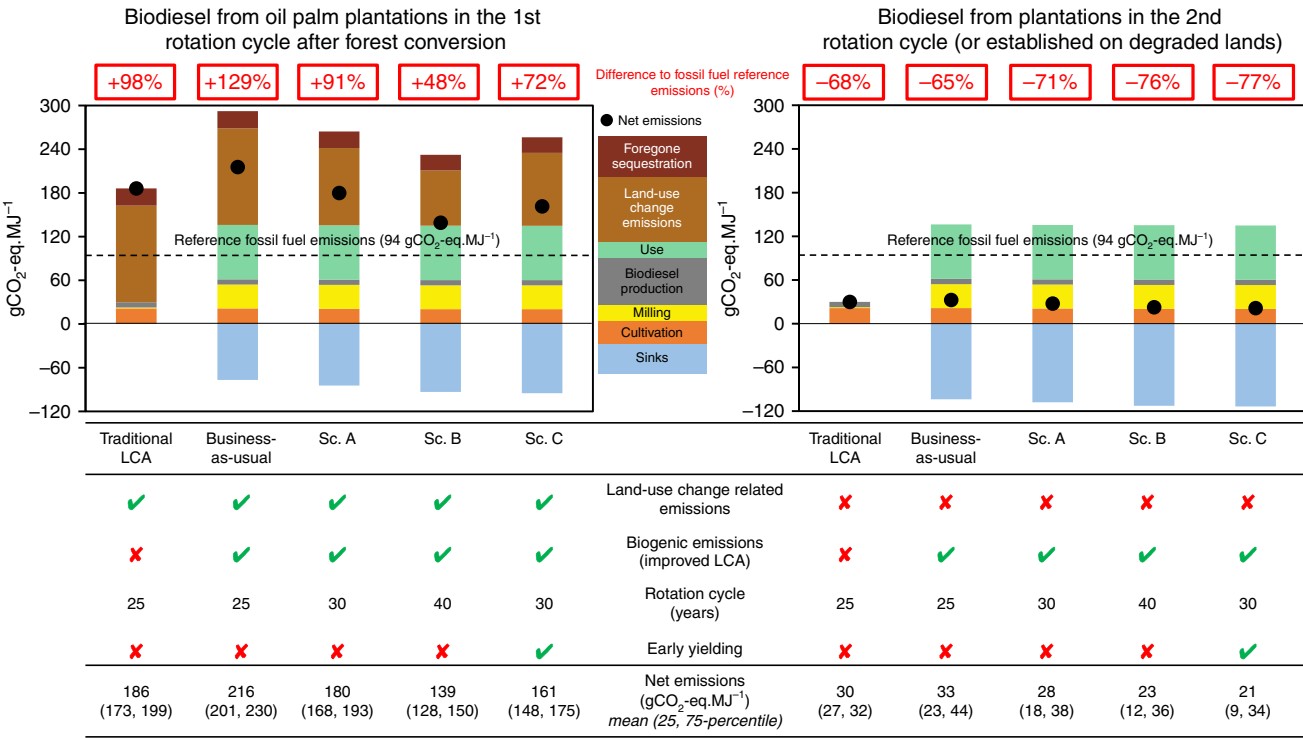

**Fig. 2 Life Cycle Assessment of palm-oil biodiesel.** Life Cycle Assessment (LCA) of palm-oil biodiesel for first (1st) and second (2nd) rotation cycles with different analysis and management scenarios. The traditional LCA follows a common LCA approach where carbon neutrality is assumed; in contrast, an enhanced LCA taking into account measured greenhouse gas (GHG) fluxes and biogenic carbon dioxide emissions from product use was applied for all other scenarios. All scenarios assume oil palm cultivation on mineral soil. The business-as-usual scenario assumes an oil palm plantation life cycle of 25 years. The simulated alternative management scenarios assume a life cycle of 30 years (scenario A), 40 years (B), and 30 years with a hypothetical, earlier-yielding oil palm variety (C). The net GHG emissions for each LCA (black dots) are the different GHG emissions over the life cycle (positive values) minus the according sinks (negative values, blue bars). Reference fossil fuel emissions are marked with a black dotted line. The lower table indicates if biogenic emissions, early-yielding varieties and land-use change emissions were considered (green check mark) or not (red cross) for each analysis, and the length of the rotation cycle. See Supplementary Fig. 3 and Supplementary Table 4 for detailed fluxes and parameters considered in all scenarios.

Carlo analysis), 98% higher than reference fossil fuel emissions (Fig. 2).

For our enhanced LCA, we included the measured GHG fluxes during the cultivation phase of a first rotation-cycle oil palm plantation on mineral soil (hereafter referred to as business-as-usual scenario; Fig. 3). This business-as-usual LCA indicates that the production of 1 MJ of biodiesel from palm oil results in total net emissions of 216 (201, 230) $gCO_2eq. MJ^{-1}$, suggesting that the traditional LCA underestimates GHG emissions by 14%. Our traditional LCA resulted in similar GHG emissions for palm oil biodiesel than previous analyses for Jambi[36], but our enhanced LCA (business-as-usual) led to 20% higher GHG emissions, confirming that the C neutrality assumption does not reflect real ecosystem emissions. Both traditional and enhanced LCA point to no GHG emissions savings compared with fossil fuel (Fig. 2). This is mainly due to high land-use change-related emissions of 156 $gCO_2$-eq. $MJ^{-1}$ (Fig. 3). Therein, land-use change emissions were calculated from C stock losses after forest conversion to oil palm plantations in the region[30], and forgone forest C sequestration[39] was derived from literature[51]. According to the enhanced LCA, oil palm cultivation, milling, biodiesel production and combustion, use of fibres and shells and production-related processes such as waste-washing water (palm oil milling effluent, POME) result in emissions of 136 $gCO_2$-eq $MJ^{-1}$. These land-use change and life cycle emissions are only partially offset by the ecosystem capture of 77 $gCO_2$-eq. $MJ^{-1}$ (sink, Figs. 2 and 3; Supplementary Table 3).

The substantial net emissions over the life cycle of a business-as-usual, first-rotation-cycle oil palm plantation mostly arise from the large $CO_2$ emissions due to forest conversion to oil palm plantations. In addition, emissions at early stages of cultivation (Supplementary Table 2) are too high to be compensated by the low photosynthetic rates of young palms (Fig. 1; Supplementary Table 1). This adds up substantial emissions over the first years after establishment, and is likely largely driven by the decomposition of soil organic carbon following the conversion of forest (Supplementary Fig. 3). The relatively high measured soil $CO_2$ fluxes in the 12-year-old plantation (Supplementary Table 2) may suggest that apart from the autotrophic respiration, there is also still ongoing soil organic carbon decomposition, more than a decade after the conversion of forest to oil palm[31,42]; in the mature, highly productive plantation, these soil emissions were, however, outbalanced by high rates of photosynthesis and thus C fixation. Our results suggest that in second-rotation-cycle oil palm plantations, i.e., after soil organic carbon has already substantially decreased 25–30 years after deforestation[31,42], soil emissions are likely much lower (Supplementary Fig. 3) and overall $CO_2$ sinks thus more balanced with $CO_2$ losses (Fig. 2), leading to a scenario that may fulfil or be closer to comply with the C neutrality assumption.

We combined NEE measured in first-rotation-cycle plantations (Fig. 1) with calibrated soil respiration functions following forest conversion (Supplementary Fig. 3) to derive LCA for biodiesel from second rotation-cycle oil palm plantations, or equivalently

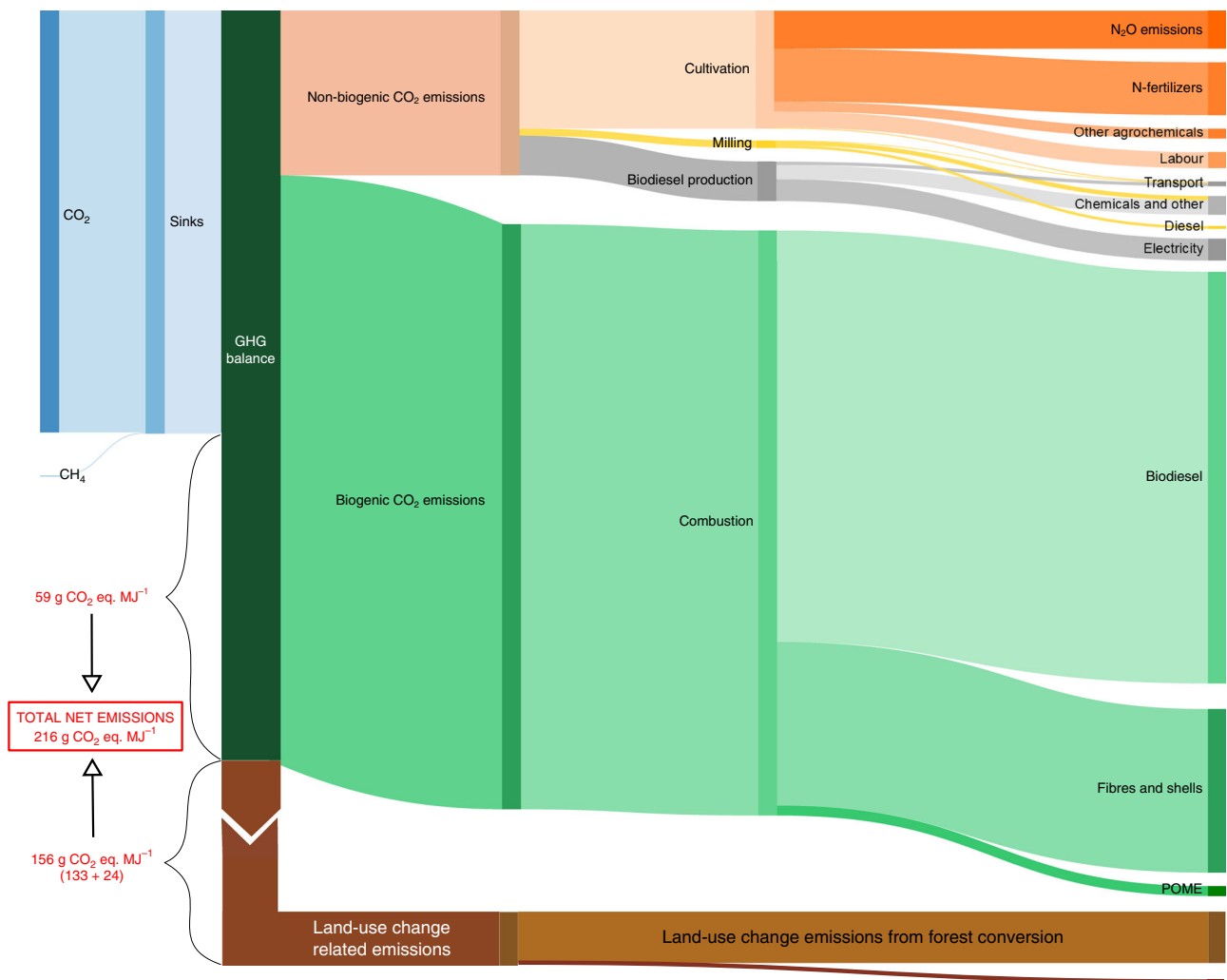

**Fig. 3 Detailed life cycle analysis for business-as-usual scenario.** Detailed life cycle analysis of palm-oil biodiesel for the business-as-usual scenario in the first-rotation cycle. Greenhouse gas (GHG) budget over the entire plantation life cycle of 25 years. Each box represents a GHG flux and box sizes are proportional to flux magnitude. The GHG balance box (dark green) comprises the GHG sinks to its left and the emissions to its right. The difference between sinks and emissions, i.e., the net GHG emissions, are indicated by brackets. Sinks (left side, blue) by the ecosystem include carbon dioxide ($CO_2$) and methane ($CH_4$) fluxes. Emissions (right side) are separated into the phase along the cycle, in which they occur, i.e., palm oil cultivation (orange), milling (yellow), biodiesel production (grey), biogenic emissions occurring during the use of the different products obtained along the system (green), land-use change emissions from forest conversion (light brown) and foregone sequestration (dark brown). All fluxes are expressed in $gCO_2$-eq. $MJ^{-1}$. Due to their large magnitude, box sizes for land-use change-related emissions are not proportional to the magnitude of the fluxes. See Supplementary Table 3 for further details on the business-as-usual scenario.

from plantations established on already degraded lands (Fig. 2). Compared with first-rotation-cycle plantations, soil emission are lower in the second rotation cycles, and no land-use change-related emissions need to be accounted for. Our results indicate that with 65% emission saving compared with fossil fuel for the business-as-usual scenario, in contrast to first-rotation-cycle plantations with no emission savings, the current EU threshold (60% savings) is met for biofuel from second-rotation-cycle oil palm plantations, and that the stricter thresholds for biofuel operations starting after 2026 (80% savings) are within reach. Accomplishing them could e.g., include optimized oil palm management scenarios.

**Management scenarios increase GHG-emission savings.** Indonesia plans to increase its palm-oil biofuel consumption to reduce dependency on imported fossil fuels[52,53]. This, together with the

continuously increasing international demand for palm oil, will likely lead to the further expansion of oil palm in the upcoming years[54]. There is thus a need for management strategies that minimize GHG emissions from the projected further expansion. The typical rotation cycle for oil palm plantations in Indonesia is 25 years, which we used as a baseline (business-as-usual)[55] even if plantations are sometimes kept for up to 30 years[56]. In other regions (e.g., Cameroon), rotation cycles are often extended to up to 40 years[57], giving room for scenarios for Indonesia with extended rotation cycles (new oil palm varieties that are currently being bred are, among other factors, also being selected for reduced height growth to facilitate harvesting[58], which will facilitate the extension of the plantation rotation cycle). In addition, early-yielding oil palm varieties are becoming increasingly available; we therefore include a scenario with a hypothetical early-yielding variety that, compared with conventional varieties, starts yielding in the third instead of the fourth year and

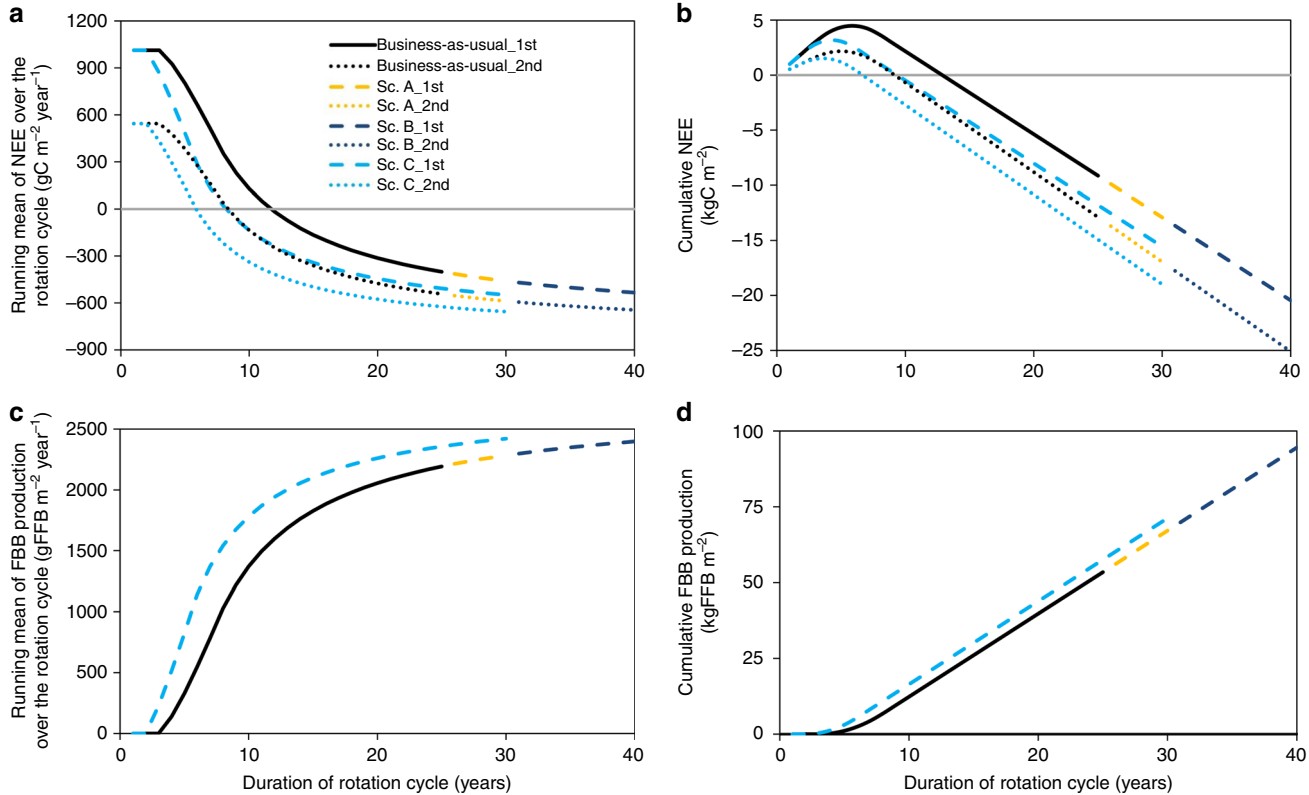

**Fig. 4 Net ecosystem exchange and yield in the rotation cycles.** Net ecosystem exchange (NEE) and yield, expressed as fresh fruit bunch (FFB) production, over the life cycles of business-as-usual and alternative management scenarios. Running annual means (left side, mean of all previous years at a given age) and cumulative values (right side) are presented for NEE (panels **a** and **b**) and for FFB production (panels **c** and **d**). All presented scenarios assume oil palm cultivation on mineral soil. The business-as-usual scenario (solid black line) assumes an oil palm plantation life cycle of 25 years. The simulated alternative management scenarios assume a life cycle of 30 years (scenario A, dashed yellow), 40 years (scenario B, dashed dark blue), and 30 years with a hypothetical, earlier-yielding oil palm variety (scenario C, dashed light blue). All scenarios for NEE are also given for second-rotation cycles (same colour codes as for first rotation cycles but with dotted lines). Yield was assumed to be constant in both rotation cycles (see Supplementary Fig. 4).

reaches maximum yield in the sixth instead of the eight year. We tested three alternative scenarios that could lead to increases in GHG-emission savings compared with the business-as-usual baseline: Scenario A, with a rotation cycle of oil palms expanded to 30 years, Scenario B, with a rotation cycle of oil palms expanded to 40 years, and Scenario C, with earlier-yielding palm varieties and a rotation cycle expanded to 30 years (Fig. 4; Supplementary Fig. 3; Supplementary Table 4). All scenarios were calculated with the enhanced LCA, i.e., including measured eco-system fluxes (Fig. 1), and GHG emissions savings were assessed for both first and second-rotation-cycle oil palm plantations (Fig. 2).

In all three alternative management scenarios (A, B, C), GHG-emission savings were larger than in the business-as-usual scenario (Fig. 2). However, because of the large land-use change-related emissions (Fig. 2; Supplementary Table 3) and high soil emissions during early stages of oil palm cultivation (Supplementary Table 2; Supplementary Fig. 2), all first-rotation cycle scenarios after forest conversion to oil palm lead to much larger emissions than reference fossil fuel. The alternative scenario B (40-year rotation-cycle), where land-use change-related emissions are divided among more years, thus logically was the scenario resulting in the lowest life cycle emissions (Fig. 2; Supplementary Table 3). This indicates that management practices aiming at increasing the rotation cycle and thus the productive period of oil palm plantations, as well as the use of oil palm varieties that reach maximum yield sooner (Fig. 4; Supplementary Table 4) can have a substantial positive effect

on GHG savings. In second rotation-cycle oil palm plantations (and equivalently in plantations established on degraded land), GHG emissions were much lower than in first-rotation-cycle plantations, with emission savings compared with fossil fuels ranging from 71 to 77% for scenarios A, B and C (business-as-usual: 65%; Fig. 2). All scenarios for palm-oil biodiesel from second-rotation cycle plantations thus comply with the current EU GHG saving requirements for biofuels, with scenario C (77% savings) already coming close to the stricter 2026 EU requirements of 80%; further GHG balance optimizations could potentially arise from new cutting-edge oil palm varieties with higher maximum yield[58].

## Discussion

Our study highlights the general relevance of using measured ecosystem GHG fluxes when performing LCA. Thus, LCA lacking data on measured GHG budgets should be reported with larger uncertainties in the final results. Further, our measurements along the cultivation cycle revealed that captures and emissions are not constant during the plantation life cycle, which affects the C neutrality assumption that is commonly used in biofuel LCA. Our improved LCA results could be used in open-source and com-mercial LCA databases to better depict oil palm plantations in future analyses. The concept could further be adapted and applied for deriving LCA of other common palm-oil products, such as comestibles or cosmetics. In a next step, such enhanced, measurement-based results could also be combined with available spatial information on land-use at larger scales[36,59], in order to

assess the impact of palm-oil biodiesel at regional and national levels.

In first-rotation-cycle oil palm plantations after forest conversion, we found large C uptake in a mature oil palm plantation, which at the ecosystem level was outbalanced by the large harvest C export, thus making the plantation a C source (Fig. 1). The young plantation was an even larger source as a consequence of its smaller photosynthetic rates, and its higher soil respiration than the mature plantation. These large differences among different stages of oil palm development (Fig. 1) suggest that the full oil palm rotation period (25–40 years) should be considered in GHG balances and LCA of palm-oil biodiesel. We show that when actual field measurements of ecosystem GHG fluxes are included in LCA, GHG-emission savings from biodiesel are reduced compared with traditional LCA, which assumes C neutrality of biogenic emissions (Figs. 2 and 3). We further show that GHG-emission savings could potentially be enhanced through optimized management, i.e., longer rotation cycles and earlier-yielding oil palm varieties (Figs. 2 and 4).

Our analyses assume that oil palm plantations were converted from forests with a relatively high above- and below-ground biomass[30]. However, in our study region conversion to oil palm also occurs from rubber agroforestry and monoculture systems with a (much) lower biomass[30,60], which would result in lower land-use change-related emissions and thus improve the GHG balance. Likewise, plantations established on degraded, non-forested land (or land with low vegetation biomass) or pastures[42] avoid such substantial biomass losses and consequent GHG emissions from land-use conversion and thus have the potential for substantial emission savings and for meeting the thresholds of the EU directives (Fig. 2). However, further measurements of emissions from second-rotation-cycle oil palm plantations or plantations on degraded lands are needed to confirm and more accurately quantify their GHG emission savings potential. A recent study, e.g., suggests that soil organic carbon stocks, after a large decrease following forest conversion to oil palm, can be partially restored by adequate management practices[61].

Our enhanced, measurement-based LCA of palm-oil biodiesel indicates that under business-as-usual management, palm-oil biodiesel from first-rotation-cycle plantations does not reach the GHG emission savings requirements (60% from reference diesel[2]) put forward in the EU directive (Fig. 2). Our business-as-usual scenario assumes oil palm cultivation on mineral soil, while GHG emissions from oil palm on organic soil (peat) are likely far higher. We did not perform a full LCA for biodiesel produced from peat soils due to incomplete data; however, our partial data series indicate seven times larger ecosystem GHG emissions from oil palm plantations on peat than from mineral soils (Fig. 1; Supplementary Table 2). This suggests that there are no GHG-emission savings compared with fossil fuel when oil palm is cultivated on organic soils. Even though the EU directive generally does allow biofuel production from cultivation on peat if certain conditions are met (e.g., no drainage), our results indicate that biodiesel from oil palm on peat likely cannot be considered a biofuel under the EU directive due to the high soil GHG emissions. Follow-up studies including a complete, measurement-based LCA of palm-oil biodiesel from peat soils should further confirm these indications.

In our improved LCA, we assumed that POME went through an anaerobic degradation process in a closed system, and that the $CH_4$ produced was recovered and burned. This assumption led to emissions of 1.8 $gCO_2eq. MJ^{-1}$ for POME, 3% of the total LCA emissions (Fig. 3). Depending on the system at the oil palm mill, these emissions could be substantially lower or higher. Theoretically, the $CH_4$ combustion could be used to produce electricity which could be fed into local grids, thus potentially reducing

POME-related emissions from palm-oil biodiesel. On the other hand, many Indonesian mills still dump POME in open lagoons[62], leading to large $CH_4$ losses to the atmosphere; this results in a ninefold increase in POME-related emissions to 16.5 $gCO_2eq. MJ^{-1}$ and would lead to substantial increases in the emissions from the business-as-usual scenario. Effective POME management is therefore a possible strategy to mitigate GHG emissions from palm-oil biodiesel.

Greenhouse gas emissions from palm-oil biodiesel from first-rotation cycle plantations are too high to lead to any emission savings, even with the implementation of optimized management scenarios (Fig. 2). However, looking ahead, our analysis shows that there is potential for emission savings for second rotation cycle plantations and for plantations established on degraded lands. Our alternative scenarios point to options for relatively quickly to implement, management-based increases in GHG emission savings, e.g., increases in rotation cycles and the use of earlier-yielding oil palm varieties. As our study was focused on GHG balance and LCA and did not cover other indicators of sustainability, the results should be interpreted in the context of further ecological, social and economic indicators of sustainable palm-oil production when designing and implementing management directives and policy on palm-oil biodiesel.

## Methods

**Study sites**. We studied two oil palm plantations in the Jambi Province, Sumatra, Indonesia, located 19 km apart and with similar climatic conditions. Average annual temperature in the area was 26.7 ± 0.2 °C, and annual precipitation was 2235 ± 385 mm (1991–2011 mean ± SD[63]), and a dry season with less than 120 mm monthly precipitation usually occurred between June and September. The first site was a 1-year-old smallholder plantation (1°50′7.6″S, 103°17′44.2″E), where we measured for about 8 months from July 2013. Palms had an upper height of up to 2 m and a trunk height of ~0.2 m during the measurement period, and still did not produce fruits. In this plantation, over 60% of the soil was covered with grasses and seasonal crops[25]. At the second site (1°41′35.0″S, 103°23′29.0″E), a state-owned plantation (PTPN VI) with a cultivation area of 2186 ha which was 12-year-old at the beginning of the study period, we measured for over 2 years from May 2014. Canopy height was 12 m at the beginning of the study period, and the oil palms were mature and produced fruits, with a mean annual yield of 26.5 $Mg ha^{-1}$ (2014–2016). Oil palms usually produce their first harvests 3–5 years after planting[55,64], and therefore in our calculations we considered yield to be produced from year 4. The PTPN plantation was fertilized with 196 $kg N ha^{-1}$ per year applied as urea during the years of study. There was hardly any understory vegetation (grasses) as herbicides (glyphosate) were routinely applied to the ground. However, the butts of pruned oil palm leaves along the trunks were densely covered with epiphytes. For additional details on both plantations, see Meijide et al.[15].

The two plantations were located on mineral soils classified as loam Acrisols[63,65]. Soil carbon (C) and nitrogen (N) contents for the 1-year-old site were 1.11 ± 0.48 and 0.08 ± 0.03%, respectively (mean ± SD). At the 12-year-old site, most of the soils were mineral (C: 1.12 ± 0.34%, N: 0.08 ± 0.02%). In addition, close to water channels nearby our study site on mineral soil and covering ~5–10% of the eddy covariance tower footprint (see next section for method description), there was an accumulation of organic matter leading to C-rich soils. While C stocks of the upper 30 cm of soil were 5.18 ± 0.66 and 4.43 ± 0.35 $kgC m^{-2}$ at the 1-year-old and on the mineral soil areas of the 12-year-old study site, they were more than five-times higher (36.17 ± 15.68 $kgC m^{-2}$) on the peat areas covered by the eddy covariance footprint in the 12-year-old plantation. The terrain was flat with only small elevation variations (±15 m, see Meijide et al.[15]) at both sites.

**Eddy covariance measurements of carbon dioxide**. The eddy covariance technique[66], which provides integrated measurements over the whole ecosystem, was used to measure net ecosystem carbon dioxide ($CO_2$) fluxes (NEE) in oil palm plantations as explained in detail in Meijide et al.[15]. Different stages of oil palm development were considered, i.e., measurements were performed in a 1-year-old plantation which was non-productive in terms of fruit yield and in a highly productive 12-year-old plantation. At both sites eddy covariance towers were built (7 and 22 m high for the 1- and 12-year-old plantations, respectively), and a sonic anemometer (Metek uSonic-3 Scientific, Elmshorn, Germany) and a fast response open-path $CO_2/H_2O$ infrared gas analyser (Li-Cor7500A, LI-COR Inc. Lincoln, USA) were installed at the top of the tower. Both instruments measured at 10 Hz. Data for the 1-year-old plantation were only available for 8 months; to estimate annual budgets, a linear extrapolation of the fluxes to 12 months was performed. In the 12-year-old plantation, fluxes were measured for 24 months; we report the according annual fluxes in our study (Supplementary Table 1). Data were

processed using the software Eddy Pro (LI-COR, Lincoln, USA). Fluxes were planar-fit coordinate rotated[67], block-averaged and corrected according to Webb–Pearman–Leuning[68]. Subsequently, data were filtered according to friction velocity, and a footprint analysis was performed following Kjlun et al.[69]. Gap-filling of meteorological and eddy covariance data were performed considering the co-variation of fluxes with meteorological variables and the temporal auto-correlation of the fluxes[70]. For further details on eddy covariance data processing at the sites, see Meijide et al.[15].

**Chamber measurements of soil greenhouse gas fluxes**. Soil GHGs fluxes ($CO_2$, $N_2O$ and $CH_4$) were measured using static vented chambers[16,21]. Chamber bases were made of made polyvinyl chloride (0.05 m$^2$), and were inserted ∼0.03 m into the soil. During sampling campaigns, these chamber bases were covered with polyethylene hoods leading to chamber heights of 0.27 m and a volume of 12 l. In each chamber, samples were removed 1, 11, 21 and 31 min after closure, using a syringe through a Luer lock. Samples were stored in previously evacuated 12-ml exetainers (Labco Limited, Lampeter, UK). Samples were transported to Göttingen (Germany), and analysed with a gas chromatograph equipped with a flame ioni-zation detector and an electron capture detector. Additional details on the sampling procedure and analysis can be found in Hassler et al.[16,21].

In the young plantation, we installed a total of nine chambers grouped into three clusters located at a maximum of 35 m from the tower and with chambers within each cluster at three different distances to the oil palm trunks (ca. 0.8, 2.5 and 5 m). We performed four measuring campaigns. In the mineral areas of the mature plantation, we selected four replicates at a distance of 50 m. Each replicate consisted of three chambers, located at 0.8, 2.8 and 4.8 m from the base of the according oil palm trunks, leading to a total of twelve chambers[21]. A total of 45 sampling campaigns were performed over 18 months. In the peat areas of the mature plantation, we installed two additional replicates (sets of three chambers leading to a total of six chambers) where we performed 37 sampling campaigns over 15 months. The installation of chambers at different distances to oil palm trunks aimed at capturing the spatial variation induced by different management practices; in smallholder oil palm plantations, for example, fertilizer is often applied directly to the trunk.

Annual fluxes were estimated based on trapezoidal interpolation between measured fluxes and the interval between sampling, and summing up interpolated fluxes for the entire year[16,21].

**Soil respiration in second-rotation cycle plantations**. Our direct measurements were restricted to oil palm plantations in their first-rotation cycle after conversion from forest. To estimate soil respiration for oil palm plantations in the second (or subsequent) rotation cycles, we calibrated a decay function from literature with the available measured data on soil respiration and C stocks. The decay function was presented by Van Straaten et al.[31], based on soil C analyses in reference forest and 17 oil palm plantations in tropical Asia, South America and Africa. It depicts relative soil C stocks compared with reference forest (100%) from 0 to 40 years after conversion to oil palm. We calibrated the curve to our study region by using the mean soil C stock measured at four study sites in the nearby Harapan rainforest as the 100% reference (5.43 ± 0.43 kgC m$^{-2}$, Guillaume et al.[30]). We confirmed that our measured soil C stocks from the 1-year-old and 12-year-old oil palm planta-tions were within the uncertainty range of the resulting decay curve before applying it to derive the equilibrium soil carbon stock under oil palm (reached 30–35 years after conversion), which was 3.26 kgC m$^{-2}$ (Supplementary Fig. 3a). Estimating the according equilibrium soil respiration was based on the assumption that the relative magnitude of soil respiration after forest conversion follows the same function over time as the previously presented relative soil C stocks (Sup-plementary Fig. 3a), and thus also reaches equilibrium 30–35 years after forest conversion. We calibrated the curve to our study region by using the measured soil respiration values of the 1-year-old oil palm plantation, as no reference forest measurements were available. Using the measured values of the 12-year-old plantation to calibrate the curve would change the results only slightly due to the small difference between measured mean and the decay curve (Supplementary Fig. 3b). From the calibrated decay function, we derived the equilibrium soil respiration under oil palm for second-rotation cycle analyses, which were 84.7 mgC m$^{-2}$ h$^{-1}$ and 825 gC m$^{-2}$ yr$^{-1}$ at hourly and annual scales, respectively.

**Net ecosystem productivity and global warming potential**. In this study, we define positive values as a flux from the ecosystem into the atmosphere, i.e., a source, whereas negative values represent a flux into the ecosystem, i.e., a sink. NEE estimates from eddy covariance measurements were combined with estimates of C removed from the field during harvest[71,72] (Cyield) to determine net ecosystem productivity (NEP$_C$[73]):

$$NEP_C = NEE + Cyield. \qquad (1)$$

NEP is positive when C is removed from the system, and negative when C is captured.

For the peat soils in the 12-year-old oil palm plantation, NEE was estimated from NEE measurements on mineral soils and soil respiration (SR) in both types of

soils:

$$NEEpeat = NEEmineral − SRmineral + SRpeat. \qquad (2)$$

For second-rotation cycles, NEE was estimated using the previously explained estimations for SR:

$$NEE2^{nd}\_rot = NEEmineral − SRmineral + SR2^{nd}\_rot. \qquad (3)$$

The contributions of $CH_4$ and $N_2O$ fluxes to the net global warming potential (GWP$_{net}$) were calculated using the most recent values for a 100-year time horizon[2,74], which are 25 and 298 for $CH_4$ and $N_2O$, respectively; they were added to the previously derived NEP to obtain the total ecosystem GWP:

$$GWP_{net} = NEP_{CO2} + GWP_{CH4} + GWP_{N2O}. \qquad (4)$$

GWP is positive when fluxes are directed away from the ecosystem (emissions).

**General assumptions on LCA**. LCA is a method to assess the potential environ-mental impacts of goods and services taking into account their whole-life cycle. It considers all inputs and outputs in terms of energy, raw materials, emissions, residues, etc. at each stage along the life cycle of a given product and assesses their potential impacts by applying different impact assessment methods. LCA is stan-dardized by the International Organization for Standardization (ISO) norms 14040:2006[75] and 14044:2006[76], which serves as the framework for LCA studies.

In traditional LCA, biofuels are considered to be C neutral, i.e., the C absorbed through photosynthesis and fixated into biomass equals the C which will be released during biodiesel combustion and any other biogenic product[38]. Only few previous studies have diverged from the assumption of C neutrality[38,50,77] due to a lack of field- measurement-based flux data. To our knowledge, no LCA of palm-oil biodiesel has applied measured GHG emissions prior to our study.

Functional unit for the LCA analysis was 1 MJ of palm-oil biodiesel. The geographical horizon of the analysis was Jambi province (Indonesia), where the palm-oil and biodiesel production were based. In order to be able to compare our results with the reference values given by the renewable energy directive of the EU[2], the GWPs published in the directive were used to calculate the midpoint impact of each GHG. The system boundaries included cultivation, milling and biodiesel production stages (Supplementary Fig. 5). For the scenarios analysed in our study, the analyses are based on time-limited data from the young and mature plantations where GHG fluxes were measured, which have been used to perform estimations for the plantation life cycle. In contrast to traditional LCA, measured $CO_2$ uptake as well as GHG emissions during different stages of the cultivation phase (see previous sections for method descriptions) were included in the analysis. $CO_2$ emissions from the combustion of the fibres, shells, the $CH_4$ collected in the mill as well as the biodiesel were also considered in the analysis. Asides from biodiesel, several co-products are obtained along the system. In order to solve this multifunctional system, allocation by energy was applied[75,76]. Primary data were used for the LCA whenever available, while data from the ecoinvent database v3[78] were used as background data. Where large divergences to ecoinvent datasets were apparent from previous work in the study region, single values were adjusted to better describe the processes in Indonesia (Supplementary Table 4). The analysis was conducted using the software SimaPro 8[79]. A sensitivity analysis based on economic revenue was conducted for the business-as-usual scenario for first-rotation oil palm plantations, to evaluate how this methodological approach would affect our results in comparison with our methodology using allocation by energy. We used biodiesel prices established by the Indonesian Ministry of Energy and Mineral Resources (7358 IDR l$^{-1}$) and crude palm oil prices for the port of Medan (7038 IDR kg$^{-1}$). Prices for glycerine, palm kernel cake and palm kernel oil were 500, 120 and 765 US \$ t$^{-1}$, respectively. Biodiesel and crude palm oil were responsible for 76.6 and 97.4% of the revenues in the production and oil extraction process, respectively. Differences in net GHG emission or in any of the processes between our LCA with allocation by energy and the analysis based on economic revenue were small (see Supplementary Table 3).

**Considerations on oil palm cultivation and harvesting**. For the LCA, oil palm cultivation begins with the first stages of plantation establishment at the nursery, where the seeds are grown until they can be transferred to the field. In our business-as-usual scenario, we assume that palms did not produce fruits until the fourth year, and that maximum productivity was reached after 8 years. During the first years, the site was treated with low amounts of agrochemicals and fertilizers (we assume fertilization rates as observed for smallholder plantations in the study area[65] for the calculations, i.e., 88 kgN ha$^{-1}$ yr$^{-1}$, 73 kgK ha$^{-1}$ yr$^{-1}$ and 38 kgP ha$^{-1}$ yr$^{-1}$). Fluxes, fertilization and herbicide application rates were considered constant during the first 3 years of cultivation, and to then linearly increase until a plan-tation age of 8 years, from whereon they remain constant. On the 8th year, when palms have reached maturity and leaf area index reached (near) maximum values[24], we also assume a plateau in NEE fluxes and yield (Supplementary Fig. 4) as well as fertilization and herbicide application rates until the end of the life cycle (i.e., 196 kgN ha$^{-1}$ yr$^{-1}$, 220 kgK ha$^{-1}$ yr$^{-1}$, 50 kgP ha$^{-1}$ yr$^{-1}$ and 2250 ml Gly-phosate ha$^{-1}$ yr$^{-1}$; see Supplementary Table 4 for fertilizer and herbicide levels along the plantation life cycle). After 25 years, although the oil palms may still be productive[55–57], a new rotation cycle is commonly initiated due to the increasing difficulty in harvesting operations with further increasing palm height. Over the

25-year life cycle, the simulated mean of fresh fruit bunches (FFB) yield was 21925 kg ha$^{-1}$ yr$^{-1}$ (including the first three non-productive years and taking into account the above-described considerations in yield development over plantation age); these values fall into the upper range of life-time FBB yield values reported in previous studies[32–34]. As we did not directly measure FFB yield in our study, the simulations were based on the data as reported by the management of the studied mature plantation (the young plantation did not produce fruits yet). The average applied fertilizers were 174.4, 190.6 and 47.6 kg ha$^{-1}$ yr$^{-1}$ for N, K and P, respectively. Glyphosate was applied at 1800 ml ha$^{-1}$ yr$^{-1}$ (Supplementary Table 4). For the LCA calculations, we assume that all fertilizers were produced in Sumatra and that only inland transport by truck occurred[27].

**Crude palm oil production in LCA.** After the FFBs were harvested in the mature plantation, the young plantation did not produce fruits yet, they were transported to the palm-oil mill located in Bunut, Sumatra (~25 km from the plantation). The mill was launched in 1996 and processes ~232,000 Mg FFB annually[80]. The FFBs first go through a sterilization process. Then the fruits are separated from the FFBs, leaving empty fruit bunches (EFBs). The EFBs are mixed with the palm oil cake (pulp left after extracting the oil from the kernel), and are applied in the plantation as natural fertilizers, thus returning the C to the soil. Before the milling process, the fruits enter in a digester and oil is obtained together with palm kernel meal[81]. To produce one kg of CPO, 5.18 kg of FFB are needed[33]. The energy required in the whole process is provided by a boiler fuelled with the fibres and shells. $CO_2$ emissions from this combustion were calculated taking into account the water content and C content of each part. Fibres and shells have 40 and 35% water content and 47.2 and 52.4% C content, respectively[33].

During the production process, waste-washing water with high organic content, also referred to as palm-oil milling effluent (POME), is generated. In the waste water treatment process POME undergoes anaerobic degradation processes in a closed system[82], during which $CH_4$ is produced, which is recovered. Hypothetically, the $CH_4$ could be used to generate electricity the surplus of which could be sold to the national grid, as it is currently being encouraged by the Indonesian government[62]. However, due to the lack of adequate electricity infrastructure in the study region, this is still not a common practice (<10% of Indonesian mills treat their POME using biogas technology[62]), and therefore we did not consider this alternative in our study. Instead, we considered that the biogas was burned, thus being released to the atmosphere as $CO_2$ emissions. If the $CH_4$ were to be directly released into the atmosphere without burning, or POME would be stored in open lagoons, where large amounts of $CH_4$ are produced, emissions from palm-oil biodiesel would increase considerably due to the high GWP of $CH_4$ (25 times higher than that of $CO_2$)[20].

**Transesterification process in LCA.** Biodiesel is produced from CPO in a plant located in the south of Sumatra. We assume a distance of 250 km for the CPO transport. The transesterification of CPO into biodiesel requires the use of methanol and sodium hydroxide and electricity as an energy source. The electricity is supplied by the Sumatran power grid; almost 90% of its electricity is produced from fossil fuels[83]. The biodiesel yield considered in this study is 95% of the CPO[33]. Besides biodiesel, the plant produces glycerol as a secondary product. Heating values of 37.5 MJ kg$^{-1}$ and 16 MJ kg$^{-1}$, respectively, were used to allocate the results[84,85]. The C content of palm-oil biodiesel amounts to 76.35%[78].

**Description of business-as-usual and alternative scenarios.** We first conducted a traditional LCA for first-rotation oil palm plantations following the common assumption of C neutrality, where any biogenic $CO_2$ emissions occurring along the life cycle are not considered as they are assumed to correspond to the $CO_2$ captured by the plant during the cultivation phase. Next, an enhanced LCA was conducted for the business-as-usual scenario (25-year rotation cycle, mineral soil, first-rotation oil palm plantation), based on the previously described field measurements of GHG fluxes in different cultivation phases. Additional biogenic $CO_2$ releases in the different phases of the life cycle were included in the LCA to ensure a closed C cycle. This included $CO_2$ emissions from the use of fibres and shells to produce the in-house energy required for the oil milling process as well as emissions from the combustion of POME and biodiesel.

Three additional scenarios were included to analyse the potential of adaptions in oil palm management to minimize GHG emissions in first-rotation oil palm plantations. We extended the plantation life cycle to 30 and 40 years (Scenarios A and B, respectively; Supplementary Table 4), maintaining the age at which palms start producing yield and reach maximum yield from the business-as-usual scenario (4 and 8 years, respectively). We analysed an additional scenario (C), where we assumed a rotation cycle of 30 years as well as the use of a (hypothetical) earlier-yielding variety compared with the other scenarios; this variety was assumed to start yielding fruits and to reach (near) maximum yield when 3- and 6-year-old, respectively (Supplementary Table 4). More details on the characteristics and applied parameters of each scenario can be found in Supplementary Fig. 4 and Supplementary Table 4. We additionally analysed these same scenarios, but for second-rotation cycle oil palm plantations, keeping all the parameters as in first-rotation oil palm plantations, but excluding land-use change-related emissions and

updating NEE along the plantation cycle (Fig. 4; Supplementary Fig. 4; Supplementary Table 4).

**Land-use change-related emissions.** We calculated $CO_2$ emissions due to C stock changes caused by land-use change during land clearance before cultivation ($E_{LU}$, kg $CO_2$-eq. ha$^{-1}$ yr$^{-1}$) following the methodology described by the European Commission[86], where:

$$E_{LU} = \left(\frac{CS_R - CS_A}{T}\right) x\,3.664. \qquad (5)$$

$CS_R$ and $CS_A$ are the C stocks per unit of area associated with the reference land use in kg C ha$^{-1}$, (land prior to conversion to bioenergy crop plantation) and the actual land (land use for bioenergy crop plantation), respectively. $T$ is the time period of land being used after conversion (i.e., plantation life cycle), and 3.664 is the conversion factor for mass carbon to mass carbon dioxide ($CO_2$).

We used measured values of C stocks within the area of study (283.5 ± 12.2 and 109.9 ± 5.5 MgC ha$^{-1}$ for forest and oil palm plantations, respectively)[30], which included C stocks in the above and below-ground vegetation as well as in the soil. The annualized emissions were calculated taking into account the net C stock (difference between C stocks of the land before conversion and C stocks of oil palm plantations). In our main analysis and results (Figs. 2 and 3), we considered that the land was converted from forest to oil palm plantation as this is the reference land-use in the region. However, many plantations in the area are converted from rubber agroforestry systems or monocultures[60], which would lead to lower emissions. In our analysis (Figs. 2 and 3), we further included the hypothetical $CO_2$ uptake that the original land use prior to the conversion (forest) would have had if it would have not been converted (referred to as 'foregone sequestration')[39]. We used NEE data from 3 years in a rainforest in Malaysia[51], which amounted to −124 ± 13 gC m$^{-2}$ yr$^{-1}$. At this site, NEE was measured with the eddy covariance technique[66], i.e., the same technique we used to measure NEE in our study. This is the closest rainforest site to our study sites for which eddy covariance data are available.

**Statistical analyses and data.** For analysis of differences in measured soil GHG emissions between 1- and 12-year-old plantations on mineral soils and between mineral and peat soils in the 12-year-old plantation, we use linear mixed effect models using the site as fixed factor (see Hassler et al.[16,21] for detailed description). We further performed the Mann–Whitney $U$ test to evaluate significant differences between the means.

We performed Monte Carlo simulations to evaluate uncertainties of our LCA for all scenarios (Fig. 2; Supplementary Fig. 6) with SimaPro 8. Herein, we used data distribution information from the ecoinvent database v3[78] for the background data and standard errors from our measurements for the measured data included in the simulations. The Ecoinvent database provides uncertainty information for all datasets based on the pedigree matrix[87,88]. The parameter uncertainties cover both the data inaccuracy and the lack of representative data. Uncertainty factors can be translated through the pedigree matrix, which includes six indicators and define different scores based on how the indicator is fulfilled. The following six categories are covered: reliability, completeness, temporal correlation, geographical correlation, further technological correlation and sample size. In our LCA, we report as uncertainties the 25 and 75 percentiles from 1000 iterations in the Monte Carlo simulations.

For NEE fluxes, we assumed a normal distribution of the data with a standard deviation of 5% of the measured value; we chose 5% because it was the maximum error obtained from a bootstrapping approach[15] of cumulative water fluxes (which have a close relationship with NEE) from the same oil palm plantation, measured with the same eddy covariance tower[15].

All further statistical analyses were carried out with R studio version 3.1.1[89]. Plotting was mainly carried out with R studio, but some figures were produced using Microsoft Excel version 14.0.6112.5000 and the LCA figure was produced using the online-tool SankeyMATIC (sankeymatic.com/). Some figures were further modified using Inkscape Project (version 0.92.4, inkscape.org).

## Data availability

Greenhouse gas flux and soil data that support the findings of this study are available at https://doi.org/10.25625/6AAOA8/GHTUYH.

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

## Acknowledgements

This study was financed by the Deutsche Forschungsgemeinschaft (DFG, German Research Foundation)— Project-ID 192626868—in the framework of the collaborative German-Indonesian research project CRC990 (subprojects A03, A04 and A05). Special thanks to our field assistants in Indonesia (Basri, Bayu and Darwis) and to Frank Tiedemann, Edgar Tunsch, Dietmar Fellert and Malte Puhan for technical assistance. We thank PTPN VI and the owner of the plantation at Pompa Air for allowing us to conduct our research at their plantation. We would also like to thank the Spanish national project GEISpain (CGL2014-52838-C2-1-R) and the DAAD (scholarship from the programme 'Research Stays for University Academics and Scientist 2018, ref. no. 91687130)' for partly financing A. Meijide during the preparation of this paper.

## Author contributions

A.M. led the writing of the paper with help from C.R. A.K. supervised the work. A.M., C.R. and A.R. performed data analysis and produced the figures. A.M., T.G., E.H., M.D.C., E.V. and A.K. contributed with data sets. T.G., A.R., E.H., C.S., A.T., T.J., M.D.C., E.V. and A.K. contributed to data interpretation, discussion and writing.

## Competing interests

The authors declare no competing interests.
