## [Peer Review File · Nature Communications]

Reviewers' comments:

Reviewer #1 (Remarks to the Author):

The manuscript entitled "Measured greenhouse gas budgets reveal limited emission savings from palm-oil biodiesel" presents an enhanced life-cycle analysis of palm oil based biodiesel. While I appreciate the detailed field-based measurements of greenhouse gas emissions at different cultivation stages and the LCA including the measurements, I consider the framing of the results incorrect or at least distorting. The first sentence of the abstract "Palm-oil biofuels are assumed to largely reduce GHG emissions compared to fossil fuels and are politically endorsed" can be used to explain my reasons for this:

First, the idea of palm oil biodiesel significantly reducing emissions compared to fossil fuels has been debunked by a number of papers 10 years ago (see e.g. Fargione et al, 2008, Germer and Sauerborn 2008, Reijnders and Huijbregts 2008, Schmidt 2010 and Wicke et al. 2008). Related to this, all LCA results up until the second to last page of the manuscript do not account for the carbon losses caused by land conversion. This comes as a surprise because the referenced EU-RED does require its inclusion and the above studies all show the importance of these emissions in the life cycle of palm oil. It is unclear why this crucial component is only briefly addressed at the end rather than incorporated in the results. Including these emissions from the start shows that conversion of primary or secondary forests can never lead to emission savings aimed at by e.g. the EU RED.

Second, political endorsement in regions that have promoted biofuels for climate change mitigation purposes (e.g. EU) has also strongly weakened (see e.g. European Commission and individual member state bills in Spring 2019). Clearly, other countries such as Indonesia and Malaysia, continue the support for palm oil biofuels, but this support does not originate in climate change mitigation but rather in (socio)economic development and independence of foreign oil imports. This framing is used throughout the manuscript to emphasize the importance of the results, but as shown here is not really valid.

However, the inclusion of field-based measurements in LCA is a clearly important step forward, and valuable for the LCA community and those people working on better understanding the emissions of palm oil. Incorporating field-based measurements can help enhance our understanding of temporal effects and differences, which may also lead to new insights for management in the future. More emphasis on this valuable addition (rather than on the pretentious novelty of the results) could actually strengthen the paper. The field-based measurements are novel, but I consider the LCA results not particularly novel –they specify what others (although with rougher LCAs) have found. Given that the land conversion emissions are not even included from the start, the presentation of traditional LCA results (e.g. Fig 2) indicating significant emission reductions compared to fossil fuels is misleading.

Alternative management scenarios are an interesting element, but particularly the scenarios with extended plantation lifetimes may not be realistic given the increasing height of palms and therefore difficulty in harvesting as well as reduced yields over time. This requires at least an explanation/reflection on these options.

The conclusions contain a very general sentence on "first-generation biofuels made from sugars or oils from food crops", which is in my opinion inappropriate given the narrow focus of the article on palm oil, rather than other first generation biofuels, while e.g. sugarcane ethanol is considered one of the cleanest biofuels.

References mentioned above:

Fargione J, Hill J, Tilman D, Polasky S, Hawthorne P. 2008. Land clearing and the biofuel carbon debt. *Science* 319: 1235– 1238.

Germer J, Sauerborn J. 2008. Estimation of the impact of oil palm plantation establishment on

greenhouse gas balance. *Environ Dev Sustain* 10: 697– 716.
Reijnders L, Huijbregts MAJ. 2008. Palm oil and the emission of carbon-based greenhouse gases. *J Clean Prod* 16: 477– 482.
Schmidt JH. 2010. Comparative life cycle assessment of rapeseed oil and palm oil. *Int J LCA* 15: 183– 197.
Wicke B, Dornburg V, Junginger M, Faaij A. 2008. Different palm oil production systems for energy purposes and their greenhouse gas implications. *Biomass Bioenerg* 28: 1322– 1337.

Reviewer #2 (Remarks to the Author):

In their paper "Measured greenhouse gas budgets reveal limited emission savings from palm-oil biodiesel" the authors integrate GHG field

measurements in oil palm plantations into a novel more elaborate form of LCA than currently in use. They clearly demonstrate that the carbon

neutrality assumption is not justified for palm-oil based agro-diesel and that real GHG-savings are probably a lot lower than previously calculated and

do not comply with EU import regulations. Additionally, the authors suggest easy to implement measures to increase GHG savings and argue that 2nd

generation oil palm plantations may have lower emissions and higher GHG savings. Results are of high scientific and political relevance, challenging

current assumptions about GHG savings of palm oil derived agrodiesel and EU import practises based on (over)simplified LCA approaches.

The paper is very well written and presented and in most parts very convincing, except the 2nd generation conclusion. The validity of the 2nd

generation conclusion depends on the assumption that soils emit lower amounts of carbon over time, an assumption which strongly depends on

(steep exponential) soil carbon decay curves. I suggest to provide evidence or plausible assumptions for the SOM decay on their sites or sites under

similar conditions and discuss this issue in the paper (see decay curve file attached).

I strongly encourage the authors to sharpen their conclusions, especially for oil palm production on organic soils, better reflecting their high quality of

field results, the novel LCA approach and the global relevance of the topic. I agree with the authors that their field data for organic soils are a bit

weaker, but the authors could follow a similar strategy as for the 1yr old plantations, for which they complement measurements with plausible field

data. This would enable the authors to complement Figure 1 with 1c = GHGs from organic soils and also calculate GHG saving scenarios and their

consequences for plantations on organic soils.

In the manuscript and the supplementary material all measurements, calculation steps and

assumptions seem plausible and transparent, but partly

could be strengthened by providing more evidence (see also detailed comments). The latter is especially important, as the paper has a high potential to

be very influential, which on the other hand may include strong comments or harsh critique e.g. from producers or policy.

Please see detailed comments and suggestions in the PDF files.

Reviewer #3 (Remarks to the Author):

The paper "Measured greenhouse gas budgets reveal limited emission savings from palm-oil biodiesel" from Meijide et al. provides an interesting life cycle assessment of palm oil-based biodiesel, based on GHG measurements of two oil palm plantations in the Jambi Province, Sumatra, Indonesia. The novelty is in measuring the GHG fluxes in the production of the fresh fruit bunches (FFB), showing that these fluxes change over time with important consequences for the net GHG balance of the biodiesel produced. These type of assessments typically require a combination of different data sources with a number of assumptions and conceptual decisions. My main comments refer to some of the decisions made in the assessment, how generalizable the approach is, and some technical questions.

I recommend publication after the authors have addressed the following comments:

1. The authors provide a comprehensive GHG balance of the FFB production, but it should be noted that the GHG fluxes should be compared with the GHG fluxes of the natural system, as if the oil palm plantation would not have been there. Now, it is assumed that the net GHG balance of the natural system is zero (not net sequestration or emission of carbon of the natural system that was before the oil palm plantation. I am not referring to the initial biomass release after clearing, but to the natural system as such. This is also called foregone sequestration. See a nice paper on this matter with focus on oil palm agriculture by Burton et al. (2017) in *Conservation Letters* (<https://doi.org/10.1111/conl.12265>).

2. The authors include allocation between co-products based on energy. This is not according the ISO14000 rules. I recommend to do this on the basis of economic revenue, particularly between glycerol and biodiesel (or follow a consequential approach as suggested in the LCA literature). Energy is in my opinion not a defensible basis for the allocation, as glycerol is not used to produce energy.

3. What are the recommendations for LCAs that require information for palm oil? It is most likely not possible to measure these fluxes., as you did. But are there lessons to be learnt from your paper, are their proxies that could be used by the LCA community to do a better job that currently done and/or are there alternatives based on other datasets, as e.g. shown by Burton et al. (2017, see above) and Lam et al. (2019: <https://doi.org/10.1016/j.scitotenv.2019.06.377>)

4. I do feel that the authors are a bit too pessimistic about the current state of the LCA literature on accounting for biogenic carbon sources in palm oil life cycles. See e.g. a recent contribution of Lam et al. (2019), but also papers from Carlson et al. (2012; 2013). I recommend that the authors discuss their findings compared to the findings in these papers.

- Carlson et al., 2012. Committed carbon emissions, deforestation, and community land conversion from oil palm plantation expansion in West Kalimantan, Indonesia. *Proc. Natl. Acad. Sci.* 109, 7559–7564.

- Carlson et al 2013. Carbon emissions from forest conversion by Kalimantan oil palm plantations. *Nat. Clim. Chang.* 3, 283–287.

Technical comments:

1. Be careful with the wording of second generation, as this term is typically used for biofuels that are produced from non-food biomass, such short rotation coppicing. I recommend that the authors reconsider their wording here.

2. P. 15: How did you arrive that the "additional emissions of 167.95 gCO₂-eq. MJ⁻¹ (conversion from forest) or 55.82 gCO₂-eq. MJ⁻¹ (conversion from agroforest)"? What are the underlying literature sources for this and what is the plantation time assumed to allocate the initial biomass loss to the biodiesel production? 20 or 30 years?

3. Be careful with reporting the numbers. You report typically 2 digits of the footprints, e.g. 29.21 g CO₂-eq on page 9. These numbers provide a picture of confidence that does not relate to the uncertainty of the numbers used. I recommend as minimum that the authors leave out the digits in the reporting of their results and consider to include an estimation of the uncertainty in their main results (not only supplementary info).

Point-by-point response to the referees' comments

Reviewers' comments:

Reviewer #1 (Remarks to the Author):

The manuscript entitled “Measured greenhouse gas budgets reveal limited emission savings from palm-oil biodiesel” presents an enhanced life-cycle analysis of palm oil based biodiesel. While I appreciate the detailed field-based measurements of greenhouse gas emissions at different cultivation stages and the LCA including the measurements, I consider the framing of the results incorrect or at least distorting. The first sentence of the abstract “Palm-oil biofuels are assumed to largely reduce GHG emissions compared to fossil fuels and are politically endorsed” can be used to explain my reasons for this:

First, the idea of palm oil biodiesel significantly reducing emissions compared to fossil fuels has been debunked by a number of papers 10 years ago (see e.g. Fargione et al, 2008, Germer and Sauerborn 2008, Reijnders and Huijbregts 2008, Schmidt 2010 and Wicke et al. 2008). Related to this, all LCA results up until the second to last page of the manuscript do not account for the carbon losses caused by land conversion. This comes as a surprise because the referenced EU-RED does require its inclusion and the above studies all show the importance of these emissions in the life cycle of palm oil. It is unclear why this crucial component is only briefly addressed at the end rather than incorporated in the results. Including these emissions from the start shows that conversion of primary or secondary forests can never lead to emission savings aimed at by e.g. the EU RED.

Second, political endorsement in regions that have promoted biofuels for climate change mitigation purposes (e.g. EU) has also strongly weakened (see e.g. European Commission and individual member state bills in Spring 2019). Clearly, other countries such as Indonesia and Malaysia, continue the support for palm oil biofuels, but this support does not originate in climate change mitigation but rather in (socio)economic development and independence of foreign oil imports. This framing is used throughout the manuscript to emphasize the importance of the results, but as shown here is not really valid.

However, the inclusion of field-based measurements in LCA is a clearly important step forward, and valuable for the LCA community and those people working on better understanding the emissions of palm oil. Incorporating field-based measurements can help enhance our understanding of temporal effects and differences, which may also lead to new insights for management in the future. More emphasis on this valuable addition (rather than on the pretentious novelty of the results) could actually strengthen the paper. The field-based measurements are novel, but I consider the LCA results not particular novel –they specify what others (although with rougher LCAs) have found. Given that the land conversion emissions are not even included from the start, the presentation of traditional LCA results (e.g. Fig 2) indicating significant emission reductions compared to fossil fuels is misleading.

Response: We thank the reviewer for the insightful comments and for acknowledging the added value of our field-measured GHG fluxes in the LCA analysis. We agree that this is the mayor strength and novelty of our study and therefore, as suggested, we have highlighted it better throughout the manuscript (in the abstract, introduction and final remarks).

We are aware of some of the controversy regarding the possible emission savings from palm-oil biodiesel, especially when plantations are established on previously forested land, and we thank the reviewer for pointing out further previous studies. We have modified the text to point out more clearly the existence of previous studies showing that palm-oil biodiesel does not reduce GHG emissions when high land-use change related emissions occur.

We are further aware of the decreasing political endorsement of palm-oil biodiesel by the EU and other countries, which reflects the findings of previous studies on limited emissions savings. However, in 2018 the EU consumed more than 7 million tons of palm oil, of which more than half were used for biodiesel, more than a third of the biodiesel produced in the EU came from palm oil, and more than a third of the biodiesel imports to the EU came from Indonesia and Malaysia (and therefore are assumed to be from palm oil; Source OILWORLD, see <https://www.transportenvironment.org/sites/te/files/publications/final%20palm%20briefing%202019.pdf>) (ref. 3). Despite the decreasing political endorsement, large amounts of palm-oil biodiesel are still being used. Following the recommendation of the reviewer, we have reworded our text, now pointing out that the use of palm-oil biodiesel is not promoted that much anymore, but that still large amounts are being used in the EU.

Regarding the valid criticism of the reviewer on the lack of land-use change emissions in our previous LCA, we now include the losses of C stored in the forest (as requested by the reviewer and in accordance with the EU RED). We have further also included the potential foregone C sequestration by the forest that is lost due to the land-use change (as requested by a further reviewer).

Some of our GHG budgets and at least the LCA results from the cultivation phase could further be applicable for LCA studies on other palm-oil based products (e.g. comestibles or cosmetics), which we now point out in the manuscript.

Alternative management scenarios are an interesting element, but particularly the scenarios with extended plantation lifetimes may not be realistic given the increasing height of palms and therefore difficulty in harvesting as well as reduced yields over time. This requires at least an explanation/reflection on these options.

Response: *We agree that most oil palm plantations are replanted after 25 years, mainly because the height of palms makes harvest difficult, but also because of potentially lowering yields. However, in the area around our study sites several plantations of more than 30 years can be found (including some owned by the same company, PTPN VI). There are reports in literature of 29 and 30 years old oil palm plantations in Sumatra (Woodham et al., 2019; ref. 56) and of more than 34 year old plantations in Cameroon (Nkongho et al., 2015; ref. 57). While plantations with such an extended life cycle may (still) be the exception, we keep our scenarios with life cycle extensions to 30 and 40 years, respectively, in order to assess easy-to-implement management measures with GHG emissions saving potential. New oil palm varieties that are currently being bred are, among other factors, also being selected for reduced height growth (to facilitate harvesting, i.e. Arolu et al, 2016; ref. 58). We have added references regarding longer plantation cycles in the text to support our scenarios.*

- Woodham C.R., Aryawan, A.A.K., Luke, S.H., Manning, P., Caliman, J.-P., Naim, M., Turner, E.C., & Slade, E.M. Effects of Replanting and Retention of Mature Oil Palm Riparian Buffers

on Ecosystem Functioning in Oil Palm Plantations. Frontiers in Forests and Global Change **2**, <https://doi.org/10.3389/ffgc.2019.00029>, (2019).

- *Nkongho, R.N., Ndjogui, T.E. & Levang, P. History of partnership between agro-industries and oil palm smallholders in Cameroon. OCL - Oilseeds and fats, Crops and Lipid* **22**, A301 (2015).
- *Arolu, I.W., Rafii, M.Y., Marjuni, M., Hanafi, M.M., Sulaiman, Z., Rahim, H.A., Kolapo, O.K., Abidin, M.I.Z, Amiruddin, M.D., Din, A.K. & Nookiah, R. Genetic variability analysis and selection of pisifera palms for commercial production of high yielding and dwarf oil palm planting materials. Industrial Crops and Products* **90**, 135-141 (2016).

The conclusions contain a very general sentence on “first-generation biofuels made from sugars or oils from food crops”, which is in my opinion inappropriate given the narrow focus of the article on palm oil, rather than other first generation biofuels, while e.g. sugarcane ethanol is considered one of the cleanest biofuels.

Response: We agree with the reviewer’s comment and have revised the Conclusions accordingly. This sentence has been removed.

References mentioned above:

Fargione J, Hill J, Tilman D, Polasky S, Hawthorne P. 2008. Land clearing and the biofuel carbon debt. *Science* 319: 1235– 1238.

Germer J, Sauerborn J. 2008. Estimation of the impact of oil palm plantation establishment on greenhouse gas balance. *Environ Dev Sustain* 10: 697– 716.

Reijnders L, Huijbregts MAJ. 2008. Palm oil and the emission of carbon-based greenhouse gases. *J Clean Prod* 16: 477– 482.

Schmidt JH. 2010. Comparative life cycle assessment of rapeseed oil and palm oil. *Int J LCA* 15: 183– 197.

Wicke B, Dornburg V, Junginger M, Faaij A. 2008. Different palm oil production systems for energy purposes and their greenhouse gas implications. *Biomass Bioenerg* 28: 1322– 1337.

Reviewer #2 (Remarks to the Author):

In their paper "Measured greenhouse gas budgets reveal limited emission savings from palm-oil biodiesel" the authors integrate GHG field measurements in oil palm plantations into a novel more elaborate form of LCA than currently in use. They clearly demonstrate that the carbon neutrality assumption is not justified for palm-oil based agro-diesel and that real GHG-savings are probably a lot lower than previously calculated and do not comply with EU import regulations. Additionally, the authors suggest easy to implement measures to increase GHG savings and argue that 2nd generation oil palm plantations may have lower emissions and higher GHG savings. Results are of high scientific and political relevance, challenging current assumptions about GHG savings of palm oil derived agrodiesel and EU import practises based on (over)simplified LCA approaches.

The paper is very well written and presented and in most parts very convincing, except the 2nd generation conclusion. The validity of the 2nd generation conclusion depends on the assumption that soils emit lower amounts of carbon over time, an assumption which strongly depends on (steep exponential) soil carbon

decay curves. I suggest to provide evidence or plausible assumptions for the SOM decay on their sites or sites under similar conditions and discuss this issue in the paper (see decay curve file attached).

I strongly encourage the authors to sharpen their conclusions, especially for oil palm production on organic soils, better reflecting their high quality of field results, the novel LCA approach and the global relevance of the topic. I agree with the authors that their field data for organic soils are a bit weaker, but the authors could follow a similar strategy as for the 1yr old plantations, for which they complement measurements with plausible field data. This would enable the authors to complement Figure 1 with 1c = GHGs from organic soils and also calculate GHG saving scenarios and their consequences for plantations on organic soils.

In the manuscript and the supplementary material all measurements, calculation steps and assumptions seem plausible and transparent, but partly could be strengthened by providing more evidence (see also detailed comments). The latter is especially important, as the paper has a high potential to be very influential, which on the other hand may include strong comments or harsh critique e.g. from producers or policy.

Please see detailed comments and suggestions in the PDF files.

Response: We thank the reviewer for the very positive evaluation of our manuscript and for the provided comments, which we believe have helped us to further improve the manuscript. We now present LCA results for palm-oil biodiesel from both first-rotation cycle oil palm plantations after forest conversion (including land-use change emissions) and second-rotation cycle plantations or plantations established on degraded land, the latter without land-use change emissions and with reduced soil carbon emissions. This was inspired by the reviewer's comment on validating and quantifying our assumptions on soil carbon emissions in first vs. second rotation-cycle plantations.

As suggested, we have added a detailed justification and reasoning for soil carbon decay after forest conversion to oil palm plantations to the Methods. We calibrated a decay function from literature with our measured data; the decay function was originally presented by Van Straaten et al. (ref. 31) based on soil C analyses in reference forest and 17 oil palm plantations in tropical Asia, South America and Africa. It depicts relative soil C stocks compared to reference forest (100%) from 0 to 40 years after conversion to oil palm. We calibrated the curve to our study region by using the mean soil C stock measured at four study sites in a nearby rainforest (Guillaume et al., ref. 30). We confirmed that our measured soil C stocks from the 1-year old and 12-year old oil palm plantations were within the uncertainty range of the resulting decay function before applying it to derive the equilibrium soil carbon stock under oil palm (reached 30-35 years after conversion). This is also depicted in a new figure (Supplementary Figure 3a; see Fig. below). In analogy to this soil carbon loss function, we further estimated the according equilibrium soil respiration over plantation age, based on the assumption that the relative magnitude of soil respiration after forest conversion follows the same function over time as the relative soil C stocks presented in Supplementary Figure 3a. Soil respiration thus also reaches equilibrium 30-35 years after forest conversion. We calibrated the curve to our study region by using the measured soil respiration values of the 1-year old oil palm plantation, as no reference forest measurements were available. The results are shown in a new figure (Supplementary Figure 3b; see Fig. below). From the calibrated decay function, we derived the equilibrium soil respiration under oil palm for the second rotation cycle LCA.

Using this equilibrium soil respiration, we estimated CO₂ fluxes along the plantation cycle for second-rotation oil palm plantations, and have run all our scenarios (traditional LCA, business-as-usual, and the 3 hypothetical scenarios with longer plantation cycles and early yielding varieties). These LCA support our conclusions that biodiesel from second rotation-cycle oil palm plantations has potential for emissions savings, which is not the case for first rotation-cycles oil palm plantations. We have updated Figure 2 to present these additional LCA for second rotation-cycle oil palm plantations (see updated Fig. 2 below):

Regarding our results for oil palm on organic soils, we have made an effort to enhance the previous presentation of the results and to sharpen the conclusions of our study regarding this issue. We followed up on the suggestion of the reviewer and added our results for peat soils for a mature oil palm plantation to Fig. 1 (see updated Fig. 1 below); as suggested by the reviewer, we completed our measurements with available data and made some assumptions to estimate net ecosystem exchange (NEE) for mature plantations on peat soils. This is now calculated based on NEE for mineral soils and soil respiration (SR) for mineral and organic soils according to the following formula (also see Methods):

$$NEE_{peat} = NEE_{min} - SR_{min} + SR_{peat}$$

As expected, our analysis shows much higher GHG emissions from peat soils than from mineral soils, with a GWP about seven-times larger. This is now explained in the results and highlighted in the conclusions. We further added a detailed description of the assumptions behind the peat estimate to the Methods. We do not have any measurements in young plantations on peat at all, where due to the high soil C contents soil respiration is expected to be very large. We saw no way of plausibly estimating fluxes for this plantation stage from existing data on mineral soil, and therefore refrain from providing a full LCA for palm-oil biodiesel from plantations on peat soils. However, the insights derived from Fig. 1, i.e. several times higher GHG emissions from mature oil palm plantations on peat than on mineral soil, should be sufficient to rule out the possibility of any emissions savings compared to fossil fuels for oil palm cultivation on organic soils. This line of argument has also been added to the manuscript text.

As further suggested by the reviewer, we have also made an effort to strengthen the methodology by providing more evidence, by making our data open access, and by explaining the data we used in more detail and by including additional references in the method descriptions. We have further addressed all of the specific comments that the reviewer provided in the separate pdf:

Specific reviewer suggestions in the PDF file by section:

Abstract

- please avoid decimals here.
Response: As suggested, we have removed the decimals in the percentages of emission savings.

Introduction

- please add "and fluxes"
Response: done

Larger GHG emissions in young plantations and on organic soils

- Please explain Diel.
Response: We have replaced the word "diel" with "diurnal".
- delete "soil"?

Response: Regarding “oil palm canopy soil”: by canopy soil, we refer to the soil lodged between the stems and leaf axils of oil palms (see Allen et al., 2018, ref 43), and not to the canopy itself. As this is very specific to oil palms, we have clarified it in the text.

- I suggest to provide an uncertainty range instead of just a mean value, taking major uncertainties into account, as you did in the supplementary material.

Response: As suggested, we have provided an uncertainty range for the GHGI. We have also updated the uncertainties in the GWP in Supp. Table 2, as we found a previous calculation error.

- please provide numbers based on your measurements and present differences between peat and mineral soils, even if they are not significantly different at a certain p level (see <https://www.nature.com/articles/d41586-019-00857-9>)

Response: We now provide numbers for emissions from peat vs. mineral soils and explain the observed differences.

- Too vague. Please quantify.

Response: The reviewer requested to quantify how much larger the emissions would be if a larger percentage of the plantation is on peat soils or if plantations are fully located on peat. We agree with the reviewer that these data would be very interesting. To make these estimations following the same procedure we used for mineral soils, in order for them to be comparable, would require ecosystem CO₂ (NEE) measurements in both young and mature plantations on peat soils. NEE fluxes are the largest component of the GHG budget, and including field NEE measurements from mineral soils and particularly those from young plantations in the LCA calculations is the main novelty of our study. Given the lack of measurements on peat, particularly at young plantation stages (where soil respiration is expected to be very large), we are thus reluctant to include full LCA for palm-oil biodiesel from plantations on peat; we also found no according estimates in the available literature. However, as described in more detail above, we followed up on the suggestion of the reviewer and added a peat soil scenario for a mature oil palm plantation to Fig. 1; it shows seven-times higher GHG emissions than from mineral soils. The assumptions and calculations behind this estimate were added to the Methods. The substantially higher emissions from oil palm plantations on peat compared to mineral soil rule out the possibility of emissions savings compared to fossil fuels for oil palm cultivation on organic soils; this insight has also been added to the manuscript text.

A further reason for not providing a full LCA for oil-palm biodiesel from peatland plantations is that the management of plantations on peat soils might be very different than observed for mineral soil (e.g. possible drainage of peatlands, machinery cannot be used in the same way as on mineral soils, fertilization levels might be different, etc.), and that we lack the required detailed information on such management procedures for running a full LCA.

Figure 1

- Well done. Excellent figure with all major fluxes. Due to the considerable differences, I strongly suggest to present systems on organic and mineral soils separately e.g. as b1 and b2.
Response: As suggested, we included fluxes from oil palm plantations on peat soils in Fig.1.

Enhanced LCA of palm-oil biodiesel reveals limited emission savings

- Throughout the paper please rethink the number of decimals. I suggest to reduce.
Response: As suggested, we have reduced the number of decimals in our LCA estimations, and left no decimals in the text. We decided to keep one decimal in Supplementary Table 3, in order to be able to see some of the smaller fluxes (e.g. emissions from transport and glyphosate are $0.2 \text{ gCO}_2\text{-eq. MJ}^{-1}$). For the ecosystem fluxes, no decimals are provided for NEE, and we kept two decimal in the annual estimates from chamber measurements as it is the typical uncertainty provided for this type of measurement (see e.g. Hassler et al., 2015, 2017; ref. 16 and 21 respectively).
- The authors are probably aware of the POME plans of the Indonesian government (see e.g. CIFOR news for a recent discussion). May be worth to address the issue briefly in the discussion or the M&M section.
Response: We are aware of the POME plans by the Indonesian government, but also that the reuse of energy from POME is still far from reality at most Indonesian mills. As the production of energy from POME is still not a common practice, we decided not to include it in our analysis. We chose an intermediate scenario, i.e. we also did not consider the most negative scenario, in which POME is stored in open lagoons, releasing CO_2 and large amounts of CH_4 to the atmosphere, which results in substantially higher GHG emissions. In our intermediate scenario, POME undergoes an anaerobic degradation that leads to CH_4 production. We assumed that the CH_4 is recovered and burned, and therefore emitted as CO_2 , which has a much lower global warming potential than CH_4 . We agree that recovering energy from POME could potentially enhance the GHG budget of palm-oil biodiesel, and have therefore included these arguments into the Discussion. We further discuss the implications of dumping POME in open lagoons and have clarified the according methods descriptions and considerations for POME use.
- Please provide error margins
Response: We have now calculated error margins for all our LCA results, which are provided in Fig. 2 and Supplementary Fig. 6. For simplification purposes we did not translate these errors into the emission savings, as with the new analysis for first- and second- rotation oil palm plantations, the percentage of emission savings for each of the different scenarios are not that relevant, and what mainly affects the results is if palm-oil biodiesel comes from first- or second-rotation cycle oil palm plantations.
- Decay curves of SOM may substantially differ between organic and mineral soils. Additionally, management strategies such as weeding, which affect soil temperature may also influence decay curves. You may want to address these issues, as they affect one of the main conclusions of your paper, i.e. that 2nd gen plantations are emitting much less CO_2 than 1st gen. In the Excel file attached I demonstrate that 2nd gen soils may emit between almost zero C and almost the same amount than 1st gen, depending on decay rates.
Response: We agree with the reviewer that different SOM decay curves may have a large impact on soil CO_2 emissions in second-rotation cycle oil palm plantations. We based our results on the decrease of soil respiration that we measured between the 1 and the 12 year old plantation on mineral soils, as well as on the soil organic carbon decay curves published in the following 2 studies:
 1. Van Straaten, O. et al. Conversion of lowland tropical forests to tree cash crop plantations loses up to one-half of stored soil organic carbon. Proceedings of the

National Academy of Sciences of the USA 112, 9956-9960 (2015). (ref. 31) See Fig. 6, showing that SOM decay levels after ca. 25 years from conversion.

2. *Quezada, J.C., Etter, A., Ghazoul, J., Buttler, A., & Guillaume, T. Carbon neutral expansion of oil palm plantations in the Neotropics. Science Advances 5, eaaw4418 (2019) (ref. 42). See Fig. 1, showing that SOC stocks constantly decreased until the beginning of the 2nd oil palm cycle, after which soil C stocks stabilized along the 2nd rotation cycle.*

As explained in detail above and in the Methods, we calibrated the soil carbon decay curves of van Straaten et al., 2015, with our soil carbon and soil respiration measurements, and observed a very good fit (see new Supplementary Fig. 3), which is also integrated into the text to support our conclusions on GHG emissions from second rotation-cycle oil palm plantations. However, we consider our conclusions on lower soil emissions from second rotation-cycle plantations only adequate for mineral soils; due to lack of key data (no soil respiration measurements from young plantations on peat, no available soil C decay curves for plantations on peat) we cannot apply our methodology to perform a full LCA for oil palm plantations on peat soils.

Figure 2

Nice figure.

- 1) May I suggest to delete the red arrows; they are not explained in the caption/legend and my impression is they are not needed to understand the figure.

Response: *done*

- 2) I would strongly prefer to see the errors (s.e) presented of suppl fig 5 here instead of just means. To avoid losing the clarity of the figure you could include them in smaller fonts in the red boxes.

Response: *We included the errors from Suppl. Fig 5 into the figure; as space was limited within the figure itself and in order not to lose the clarity of the figure, we provide them as uncertainty ranges in an additional row in the table located below the figure.*

- 3) Oil palm and other production systems on tropical organic soils are a major concern due to the large amounts of carbon stored and lost. Unclear why you focus on mineral soils and miss the opportunity to address organic soils here (I am aware of your missing data argument below). I'd be happy to see additional scenarios being addressed in this paper or at least in the supplementary material.

Response: *As explained in detail above, we did not see a sufficiently strong data basis to plausibly back assumptions regarding GHG emission from young oil palm plantations on peat soils and thus opted to not present a full LCA for palm-oil biodiesel from plantations on peat due to the high and unknown associated uncertainties. We however included additional scenarios for second-rotation oil palm plantations.*

Reconciling oil palm management and GHG budgets

- Forests and agroforestry systems cover a wide range of carbon stocks. Please provide references for the C stocks and present plausible ranges instead of means.

Response: Data of carbon stocks from forest and agroforestry systems were obtained from Guillaume, T. et al., Carbon costs and benefits of Indonesian rainforest conversion to plantations. *Nature Communications* **9**, 2388 (2018). (ref. 30).

We have now clarified in the main text that we used this data, which were measured in the same region where our study took place. These data included ranges based on the errors provided in the above mentioned paper. Land-use change emissions are the largest component of the LCA analysis and therefore, as requested by a further reviewer, we have included these emissions in our main results and in Figures 2 and 3. Therefore, instead of reporting the uncertainties associated with these particular emissions, the uncertainties in the C stocks were included in our Monte Carlo simulations (standard errors of the C stocks in forest and oil palm plantations) and the uncertainties are provided for the net emissions for each of the scenarios (see Supplementary Fig. 6 – below- and Fig. 2). We have now only calculated land-use change emissions for conversion from forest to oil palm, as forest is the reference land use in the area. However, we indicated that land-use change emissions will be lower if conversion is from agroforestry systems or rubber monocultures. We included a section in the Methods explaining how the land-use change emissions were calculated.

- Does the number of digits really reflect the precision of your approach? My impression is that in most sections you could reduce the decimals 1-2 without losing precision.

Response: As explained above, we reduced the number of decimals for LCA analysis.

- Conclusions for production systems on peat lands remain (too) vague. One way out could be to complement the almost complete dataset with plausible literature values or measurements (comparable to the strategy you applied for Ch4 in 1-yr plantations). Doing so would enable you to argue with numbers or ranges, strengthening the argument of C losses exceeding C savings when planting on peat soils.

Response: As the results regarding palm-oil biodiesel from plantations on peat soil are among the least strong finding of our study, and because of the above explained reasons on the lack of data, we decided to keep the focus of the relatively short conclusion on aspects based on hard data, i.e. different LCA-derived emissions savings for first vs. second rotation cycle plantations and the effects of optimized management scenarios.

Methods

- Please explain: Chambers were stored...

Response: We thank the reviewer for checking the methods so carefully. This was a mistake and has now being modified to “samples were stored...”.

- Please provide the numbers here.

Response: We have clarified the fertilization levels that we used along the life cycle of palm oil cultivation in the text and, as suggested, extended Supplementary Table 4 with the detailed values of yield, N-, P-, K-fertilizers and glyphosate for our scenarios.

Supp. Fig. 5

- Please explain the types of uncertainties presented in this figure.

Response: We now explain in detail in the Methods how the uncertainties were calculated, and also included key information in the figure caption.

Supp. Table 2

- Please check No. of references (Hassler , Meijide)

Response: We thank the reviewer for pointing this out; we have corrected the references in the table.

Reviewer #3 (Remarks to the Author):

The paper “Measured greenhouse gas budgets reveal limited emission savings from palm-oil biodiesel” from Meijide et al. provides an interesting life cycle assessment of palm oil-based biodiesel, based on GHG measurements of two oil palm plantations in the Jambi Province, Sumatra, Indonesia. The novelty is in measuring the GHG fluxes in the production of the fresh fruit bunches (FFB), showing that these fluxes change over time with important consequences for the net GHG balance of the biodiesel produced. These type of assessments typically require a combination of different data sources with a number of assumptions and conceptual decisions. My main comments refer to some of the decisions made in the assessment, how generalizable the approach is, and some technical questions.

I recommend publication after the authors have addressed the following comments:

1. The authors provide a comprehensive GHG balance of the FFB production, but it should be noted that the GHG fluxes should be compared with the GHG fluxes of the natural system, as if the oil palm plantation would not have been there. Now, it is assumed that the net GHG balance of the natural system is zero (not net sequestration or emission of carbon of the natural system that was before the oil palm plantation. I am not referring to the initial biomass release after clearing, but to the natural system as such. This is also called foregone sequestration. See a nice paper on this matter with focus on oil palm agriculture by Burton et al. (2017) in Conservation Letters (<https://doi.org/10.1111/conl.12265>).

Response: We thank the reviewer for the positive evaluation of our research and for the provided comments, which we believe have helped us to further strengthen the manuscript. We agree with the reviewer that a more exhaustive analysis should include land-use change related emissions, i.e. direct emissions from forest conversion and also foregone sequestration of the natural system. We have adjusted our LCA for first rotation-cycle oil palm plantations accordingly for all scenarios (Fig. 2 and 3); in contrast, for the added analysis of second rotation-cycle plantations (or plantations established on degraded land), the land-use change related emissions are not relevant. Foregone sequestration was estimated by Burton et al., 2017 by the change in aboveground biomass. We consider that using Net Ecosystem Exchange (NEE) data would be a more adequate approach to estimate foregone sequestration, as this also includes soil respiration. Additionally, it is more consistent with the approach we used for oil palm plantations. Due to a lack of measurement-based NEE data from our study region, we used NEE data from a forest in Malaysia (Pasoh Forest Reserve) to determine the net foregone sequestration in oil palm plantations (hypothetical sequestration by the forest that did not happen due to land-use change). The Pasoh site is the closest (and probably only) forest site to Jambi that has published multiple-year eddy covariance data from tropical forest in the region (see Kosugi et al., 2008, CO₂ exchange of a tropical rainforest at Pasoh in Peninsular Malaysia, Agricultural and Forest Meteorology 148, 439-452: ref. 51).

2. The authors include allocation between co-products based on energy. This is not according the ISO14000 rules. I recommend to do this on the basis of economic revenue, particularly between glycerol

and biodiesel (or follow a consequential approach as suggested in the LCA literature). Energy is in my opinion not a defensible basis for the allocation, as glycerol is not used to produce energy.

Response: As the reviewer has highlighted, allocation is not the first procedure proposed by the ISO 14040 and 14044: 2006 rules. ISO recommends to generally avoid allocation methods by subdividing the system or by extending the system boundaries. Only where this is not possible should allocation procedures be applied. We fully agree with the reviewer that the most realistic basis for allocation between biodiesel and glycerol is their economic revenue. However, all biofuels that aim at complying with the EU-RED should declare GHG emissions using an allocation procedure based on energy, with the probable objective of minimizing the effect of volatile markets on the results. As we interpret our results in the context of GHG emissions saving as required by the EU, we decided to stick with the approach as set forwards in the EU-RED for the presentation of our main results. However, as this choice may lead to different results, we have included a sensitivity analysis based on economic revenue to the business-as-usual for first-rotation oil palm plantations scenario into the manuscript (Supplementary Table 3 and methods). This analysis did not show significant differences with the one performed using allocation by energy (215.6 vs 207.6 g CO₂-eq. MJ⁻¹ for allocation by energy and analysis by economic revenue respectively, see Supplementary Table 3 for additional details).

3. What are the recommendations for LCAs that require information for palm oil? It is most likely not possible to measure these fluxes., as you did. But are there lessons to be learnt from your paper, are their proxies that could be used by the LCA community to do a better job that currently done and/or are there alternatives based on other datasets, as e.g. shown by Burton et al. (2017, see above) and Lam et al. (2019: <https://doi.org/10.1016/j.scitotenv.2019.06.377>)

Response: Lam et al., 2019 conducted an extensive analysis of GHG footprints of palm oil production in Indonesia using high resolution spatial information to determine land use and land cover types, and therefore their associated emissions. They highlighted the large variability in GHG footprints of CPO production, ranging from 0.7 to 26 tCO₂-eq. tCPO⁻¹ depending on the region. Using information from the MoF map (Greenpeace, 2015), they estimated the GHG footprint for plantations in Jambi, the area in which our study took place, where emissions amounted to tCO₂-eq. tCPO⁻¹. We compared this value with our results, by adding to their emissions those from the downstream processes and applied the economic allocation factor. We also subtracted from our results the emissions due to carbon foregone sequestration, since they were not included in the analysis by Lam et al. Lam's GHG footprint for Jambi was very similar to our traditional LCA (155 vs 152 gCO₂-eq. MJ⁻¹ respectively). However, our improved LCA (business-as-usual) was 20% higher than the values estimated by Lam et al. (185 vs. 155 gCO₂-eq. MJ⁻¹ respectively), confirming that assuming carbon neutrality leads to an underestimation of GHG emissions. The use of detailed spatial information to determine changes in land use and land cover are essential for accounting for the GHG footprint of palm oil products, but this approach, not considering ecosystem measurements, still underestimates the emissions compared to our results. Where researchers cannot provide direct measurements of GHG at different cultivation stages to comprehensively assess biogenic emissions from cultivation, they should add least be aware (and point out) these additional uncertainties. Ideally, spatial data should be combined with field measurements to improve the GHG estimations. We recommend measuring ecosystem GHGs in second-rotation oil palm plantations to confirm our estimations. If our estimations are confirmed by field measurements, current life cycle inventories for oil palm plantations should be updated including information on GHG fluxes for first and second rotation cycles. We have added this reasoning in the Discussion.

An incorporation of the insights derived from our results into current life cycle inventory datasets is partially difficult, as those typically report aggregated input and output values over the plantation life cycle. However, future studies could correct such aggregated values for differences in emissions at different cultivation stages (instead of assuming constant emissions over the life cycle) based on data-driven studies such as the one presented here for oil palm; they should further adjust for the actual rotation cycle length. Additionally, the land-use spatial and historical information collected in tools such as the MoF map used by Lam et al., could be used to identify first and second rotation cycles. Our results could then be used to re-estimate the actual GHG emissions from oil palm plantations and all derived products. We added a brief outlook for future LCA studies summarizing these thoughts to the manuscript text.

4. I do feel that the authors are a bit too pessimistic about the current state of the LCA literature on accounting for biogenic carbon sources in palm oil life cycles. See e.g. a recent contribution of Lam et al. (2019), but also papers from Carlson et al. (2012; 2013). I recommend that the authors discuss their findings compared to the findings in these papers.

- Carlson et al., 2012. Committed carbon emissions, deforestation, and community land conversion from oil palm plantation expansion in West Kalimantan, Indonesia. *Proc. Natl. Acad. Sci.* 109, 7559–7564.

- Carlson et al 2013. Carbon emissions from forest conversion by Kalimantan oil palm plantations. *Nat. Clim. Chang.* 3, 283–287.

***Response:** We appreciate the additional literature provided by the reviewer and agree that several authors had previously already described high carbon fluxes due to deforestation and land conversion to oil palm plantations, which we have taken up in our discussions and now incorporated land-use change emissions in our analysis. Both Lam et al., 2019 and Carlson et al. (2012, 2013) use spatial information on land-cover for their estimations of carbon losses due to deforestation, which we now mention in the discussion. Regarding the lack of LCA literature on accounting for biogenic carbon sources in palm oil life cycles, Lam et al., account for emissions during the cultivation phase, but those are limited to emissions due to fertilizer application, and they still assume carbon neutrality. We now compare our results to their estimations for Jambi, where our study took place, and as explained above, obtained very similar results for our traditional LCA. Carlson et al. (2012, 2013) consider biogenic carbon sources in palm oil life cycles by evaluating changes in aboveground biomass. Unfortunately, these results are not incorporated into LCAs, so we cannot make direct comparisons, but we included these studies in our introduction.*

Technical comments:

1. Be careful with the wording of second generation, as this term is typically used for biofuels that are produced from non-food biomass, such short rotation coppicing. I recommend that the authors reconsider their wording here.

***Response:** As suggested, we have replaced the wording of “first and second generation plantations” to “first and second rotation-cycle plantations”.*

2. P. 15: How did you arrive that the “additional emissions of 167.95 gCO₂-eq. MJ⁻¹ (conversion from forest) or 55.82 gCO₂-eq. MJ⁻¹ (conversion from agroforest)”? What are the underlying literature sources

for this and what is the plantation time assumed to allocate the initial biomass loss to the biodiesel production? 20 or 30 years?

Response: The emissions due to land-use change have been calculated using values provided by Guillaume et al., 2017, who measured carbon stocks in forest and agroforests (and at our oil palm sites) within the same region and with consistent methodology. The respective differences between reference and oil palm carbon stocks are allocated over 25 years for the traditional LCA and the business-as-usual scenario. For the alternative management scenarios, the carbon losses are allocated over 30 years (Sc. A and C) and 40 years (Sc. B), respectively. As requested by another reviewer, the land-use change related emissions were added for all LCA for first rotation-cycle oil palm plantations after forest conversion (see example in Fig. 3); in contrast, they are not considered for second rotation-cycle plantations or plantations established on degraded land (Fig. 2). A detailed description of the assumptions and calculations on land-use change related emissions including references has been added to the Methods.

3. Be careful with reporting the numbers. You report typically 2 digits of the footprints, e.g. 29.21 g CO₂-eq on page 9. These numbers provide a picture of confidence that does not relate to the uncertainty of the numbers used. I recommend as minimum that the authors leave out the digits in the reporting of their results and consider to include an estimation of the uncertainty in their main results (not only supplementary info).

Response: As suggested by the reviewer, we reduced the number of digits in our LCA estimations. We decided to keep one decimal in Supplementary Table 3, in order to be able to see some of the smaller fluxes (e.g. emissions from transport and glyphosate are 0.2 gCO₂-eq. MJ⁻¹). However, we did remove the two digits in the main text and results. As measures of uncertainty, we included information on percentiles 25 and 75 of the LCA estimates in Figure 2 (see Methods for details), and also included these ranges in the manuscript text.

REVIEWERS' COMMENTS:

Reviewer #2 (Remarks to the Author):

Dear authors,
you did an excellent job in improving the manuscript, including addressing my concerns and suggestions both in terms of methodological issues (e.g. soil carbon decay; uncertainties) and conclusions (e.g. clearer addressing 1st and 2nd rotation and peat soil aspects).
I hope to see this paper published asap.
Best wishes
Joerg Priess

I am responding to your add on review request for the Mejjide et al. paper focusing on the responses to reviewer 1:

My impression is that the authors have done a great job again. Seems they realized that in the 1st version they missed an important part of the C-emissions from LUCC discussion, or addressed it too briefly, although the debate is key to the paper(, provoking a harsh and substantial comment by reviewer 1).

I think their responses are valid. While indeed the overoptimistic and naive bioenergy hype in Europe is over, we still import large amounts of palm oil from SE Asia. The new version of the paper better explains the current status and near future options, contributing new insights under which conditions these production systems or imports would make sense - more sense - no sense at all from an emission saving perspective.

To summarise:

Yes, I think they addressed the issues raised by reviewer 1, including the most critical one saying "hey you missed an important aspect of the biofuel debate"

Reviewer #3 (Remarks to the Author):

I reread the resubmitted paper "Measured greenhouse gas budgets challenge emission savings from palm-oil biodiesel" from Mejjide et al. with great pleasure. The reviewers did a great job by (1) adding the foreground sequestration component to the life cycle balance, (2) including a monetary allocation scenario to the SI, (3) providing more background information on the calculation procedure, (4) having a more extensive comparison with previous work, and (5) including some further recommendations for the LCA community. I recommend the paper for publication in Nature Communications.